# Differences in water and vapor transport through angstrom-scale pores in atomically thin membranes

Peifu Cheng [1], Francesco Fornasiero [2], Melinda L. Jue [2], Wonhee Ko [3], An-Ping Li [3], Juan Carlos Idrobo [3,7], Michael S. H. Boutilier [4] & Piran R. Kidambi [1,5,6] ✉

The transport of water through nanoscale capillaries/pores plays a prominent role in biology, ionic/molecular separations, water treatment and protective applications. However, the mechanisms of water and vapor transport through nanoscale confinements remain to be fully understood. Angstrom-scale pores (~2.8–6.6 Å) introduced into the atomically thin graphene lattice represent ideal model systems to probe water transport at the molecular-length scale with short pores (aspect ratio ~1–1.9) i.e., pore diameters approach the pore length (~3.4 Å) at the theoretical limit of material thickness. Here, we report on orders of magnitude differences (~80×) between transport of water vapor (~44.2–52.4 g m⁻² day⁻¹ Pa⁻¹) and liquid water (0.6–2 g m⁻² day⁻¹ Pa⁻¹) through nanopores (~2.8–6.6 Å in diameter) in monolayer graphene and rationalize this difference via a flow resistance model in which liquid water permeation occurs near the continuum regime whereas water vapor transport occurs in the free molecular flow regime. We demonstrate centimeter-scale atomically thin graphene membranes with up to an order of magnitude higher water vapor transport rate (~5.4–6.1 × 10⁴ g m⁻² day⁻¹) than most commercially available ultra-breathable protective materials while effectively blocking even sub-nanometer (>0.66 nm) model ions/molecules.

The transport of water through nanoscale pores/channels/capillaries/slits is central to several natural/biological processes e.g., water transport through cell membranes, nephrons[1], and aquaporin/protein channels[2–4], as well as large-scale engineered systems such as reverse osmosis membranes for desalination, water purification, and treatment[5–8]. The distinctly different phase behavior[9–13] and transport characteristics of water within nanoscale confinements[12–18] in comparison to bulk water has attracted significant interest across scientific disciplines. For example, theoretical and experimental investigations of water transport through nanopores in material systems such as thin-film composite (TFC) membranes[5–8], carbon nanotubes (CNTs)[19–21], zeolitic imidazolate frameworks (ZIFs)[22], bio-mimetic channels[23–28], carbon nanomembranes (CNMs)[18], two-dimensional (2D) capillary devices[16,17,29], 2D membranes[30–34], clay interlayers[35], among others have revealed new phenomena including unexpectedly fast flow[36], large slip lengths[16,37], and single-file movement of water molecules[21]. Yet, a

[1]Department of Chemical and Biomolecular Engineering, Vanderbilt University, Nashville, TN 37212, USA. [2]Physical and Life Sciences, Lawrence Livermore National Laboratory, Livermore, CA 94550, USA. [3]Center for Nanophase Materials Sciences, Oak Ridge National Laboratory, Oak Ridge, TN 37831, USA. [4]Department of Chemical and Biochemical Engineering, Western University, London, ON N6A 5B9, Canada. [5]Department of Mechanical Engineering, Vanderbilt University, Nashville, TN 37212, USA. [6]Vanderbilt Institute of Nanoscale Sciences and Engineering, Vanderbilt University, Nashville, TN 37212, USA. [7]Present address: Materials Science and Engineering Department, University of Washington, Seattle, WA 98195, USA. ✉e-mail: piran.kidambi@vanderbilt.edu

comprehensive understanding of the mechanisms of water and water vapor transport through nanoscale pores and capillaries remains elusive.

Atomically thin 2D materials such as monolayer graphene represent unique model systems to probe transport phenomena at the molecular length scale[38,39]. Theoretical calculations by Suk and Aluru[33] initially predicted high water fluxes through ~0.75–2.75 nm wide nanopores in graphene along with a single-file water structure. Cohen-Tanugi et al.[15] also computed high water fluxes ~10–130 L cm⁻² day⁻¹ MPa⁻¹ (~100–1300 g m⁻² day⁻¹ Pa⁻¹, surpassing conventional desalination membranes, ~0.24–2.88 g m⁻² day⁻¹ Pa⁻¹, see Supplementary Table 1)[7] through graphene nanopores ~0.15–0.89 nm in diameter (pore areas 1.5–62 Å²) while effectively rejecting salt (NaCl, hydrated diameter ~0.66–0.72 nm)[38].

Celebi et al.[40] experimentally confirmed high permeance of water ~2.7 × 10⁻⁸ m³ m⁻² s⁻¹ Pa⁻¹ (~2300 g m⁻² day⁻¹, for ~50 nm pores) and water vapor (~5 × 10⁶ g m⁻² day⁻¹ for ~400 nm pores) through few-nanometer to micron-scale pores (~7.6–1000 nm) in stacked bilayer graphene membranes and noted that, with only one side of the graphene membrane wetted, capillarity prevented the permeance of water even under few bars of applied pressure, but water vapor readily transported. In contrast, Surwade et al.[30] reported rapid water transport ~10⁶ g m⁻² s⁻¹ (6 × 10⁶ g m⁻² s⁻¹ atm⁻¹, ~5 × 10⁶ g m⁻² day⁻¹ Pa⁻¹, 40 °C, using pressure difference as the driving force and only one side of the membrane wetted) and ~70 g m⁻² s⁻¹ atm⁻¹ (~60 g m⁻² day⁻¹ Pa⁻¹, using osmotic pressure as the driving force) through nanoscale pores in micron-scale monolayer graphene while rejecting ~100% of salt (KCl, NaCl, or LiCl, hydrated diameter ~0.66–0.76 nm)[38]. However, the origins of the orders of magnitude discrepancy remain unclear. Yang et al.[31] reported much lower water permeance ~20 L m⁻² h⁻¹ bar⁻¹ (~5 g m⁻² day⁻¹ Pa⁻¹, salt rejection >97%) via forward osmosis (FO) as well as hydraulic permeability ~97.6 L m⁻² h⁻¹ bar⁻¹ (~23 g m⁻² day⁻¹ Pa⁻¹, salt rejection >86%) via reverse osmosis (RO), respectively, for nanopores etched into monolayer graphene supported on a carbon-nanotube mesh (GNM/SWNT). However, they did not probe water vapor transport and hence differences in transport characteristics remain elusive[31]. In this context, rapid water permeation ~1.1 × 10⁻⁴ mol m⁻² s⁻¹ Pa⁻¹ (~170 g m⁻² day⁻¹ Pa⁻¹) across carbon nanomembranes (CNMs) with sub-nanometer channels has also been reported[18], but the researchers were not able to unambiguously conclude if this was liquid water or water vapor transport. These observations raise intriguing questions on the mechanisms and differences in the transport of water molecules through sub-nanometer scale pores/channels.

The introduction of angstrom-scale defects into the graphene lattice allows for the creation of nanopores in an atomically thin membrane[32,41,42], where the pore length ~0.34 nm represents the theoretical minimum material thickness and approaches the molecular length scale of water ~0.28 nm (van der Waals diameter)[38]. In particular, nanopores ~2.8–6.6 Å in the graphene lattice represent ideal model systems to probe water transport, since both the diameter and the length of the pore approach molecular length scales of water, i.e., nanopore aspect ratio (diameter/length) ~1–1.9. Here, we report on orders of magnitude differences in transport rates of water and water vapor through angstrom-scale pores ~2.8–6.6 Å created in centimeter-scale monolayer graphene membranes, which emanate from differences in the flow regimes during permeation. Specifically, liquid water permeation occurs near the continuum regime, while water vapor transport occurs in the free molecular flow regime. We leverage these insights to realize centimeter-scale graphene membranes that can effectively block sub-nanometer (>0.66 nm) model molecules while maintaining an order of magnitude higher water vapor transport rate (~5.4–6.1 × 10⁴ g m⁻² day⁻¹) compared to commercially available ultra-breathable protective materials (including those with ~210 nm pores), emphasizing their potential for next-generation breathable and protective materials against chemical/biological agents.

## Results and discussion

To probe water vapor and liquid water transport across angstrom-scale pores, we fabricate centimeter-scale atomically thin graphene membranes (see schematic in Fig. 1A and methods section) by transferring as-synthesized nanoporous graphene (NG)[42] to polycarbonate track-etched (PCTE) support (see the optical image in Fig. 1B)[32,43], followed by UV/ozone etching to increase nanopore density[32,44,45], and selectively sealing large nanopores (>0.5 nm) and/or any tears via size-selective interfacial polymerization (IP)[32,43].

Scanning electron microscopy (SEM) images (Fig. 1C) reveal successful graphene transfer onto the PCTE support[43], with graphene coverage (≥96%) indicated by the fractional ethanol leakage ((PCTE + NG)/PCTE ~4%) through the as-synthesized nanoporous graphene (NG) transferred to PCTE support (Supplementary Fig. 1). Atomic resolution scanning transmission electron microscopy (STEM) images (Fig. 1D, E) confirm the introduction of a high density of defects that manifest as nanopores in the UV/ozone treated graphene lattice and the resulting pore size distribution shows that most defects are <1 nm, with few nanopores >1 nm and an overall nanopore density ~5.3 × 10¹² cm⁻² (Fig. 1F)[32]. The presence of a D peak in the Raman spectrum (Fig. 1G) confirms the existence of intrinsic defects in the as-synthesized nanoporous graphene (NG) as well as an increase in defects after UV/ozone etch[32,46,47]. Scanning tunneling microscopy (STM) (Fig. 1H) performed directly on the graphene on Cu further confirms the existence of nanopores (bright defect marked by the circle) in the graphene lattice[48,49].

The size-selective IP process leverages steric hindrance to seal only large defects (>0.5 nm) and tears in the fabricated graphene membranes (GMs, i.e., PCTE + NG + UV/ozone + IP) while preserving defects[50] <0.5 nm as confirmed via water transport (Fig. 2) as well as diffusion-driven transport (see the set-up in Fig. 2D and Methods section) measured using solutes including KCl (salt, hydrated diameter of K⁺ ~0.662 nm and Cl⁻ ~0.664 nm)[38], NaCl (salt, hydrated diameter of Na⁺ ~0.716 nm)[38], L-tryptophan (L-Tr, amino acid, ~0.7–0.9 nm, 204 Da)[51] and Vitamin B12 (B12, vitamin, ~1–1.5 nm, 1355 Da)[51]. Compared with PCTE + NG (without IP, see Fig. 1I), the GM (1–4) after IP shows significantly reduced normalized diffusive flux for KCl (<4%), NaCl (<4%), L-Tr (~1%), and B12 (<0.5%), respectively, indicating that the majority of defects >0.66 nm and tears have been effectively sealed. Diffusive transport measurements across several GMs fabricated showed similar results (Fig. 1I), confirming the reliability and reproducibility of the fabrication process.

Next, we proceed to evaluate water vapor and liquid water transport through the same GMs (see methods section) to facilitate a direct and effective comparison. For water vapor transport, we establish a purely diffusive steady-state transport by ensuring $\Delta P = 0$ across the GMs mounted in a cross-flow dynamic moisture permeation cell (DMPC, Fig. 2A)[52,53]. At 30 °C, the GMs exhibit a water vapor transmission rate (WVTR) of ~54,000 g m⁻² day⁻¹ (based on the graphene area within the PCTE pores) at relative humidity ($\overline{RH}$) = 30% and ~61,000 g m⁻² day⁻¹ for $\overline{RH}$ = 40%, respectively (see Fig. 2B), i.e., a nearly constant water vapor transport rate regardless of environmental RH (unlike polymers adsorbing water e.g., Gore-Tex which needs a certain level of hydration for achieving good WVTR values), which is an important advantage for breathable and protective materials[54,55]. We note the WVTRs are ~35 times higher than the generally accepted US military guideline for water vapor breathability in a protective garment (1500–2000 g m⁻² day⁻¹)[56], and an order of magnitude higher than most commercial breathable materials (1440–6900 g m⁻² day⁻¹, Fig. 2C)[54]. Notably, the GMs achieve these high WVTRs despite having significantly smaller pores (<0.66 nm) than most commercially available breathable materials (e.g., ePTFE with mean pore size ~210 nm)[57]. Furthermore, even by accounting for just 9.4% of the area of GMs (as measured, see Supplementary Fig. 2), WVTR up to ~5100 g m⁻² day⁻¹ can still be readily achieved, which is >3× the WVTR value required for

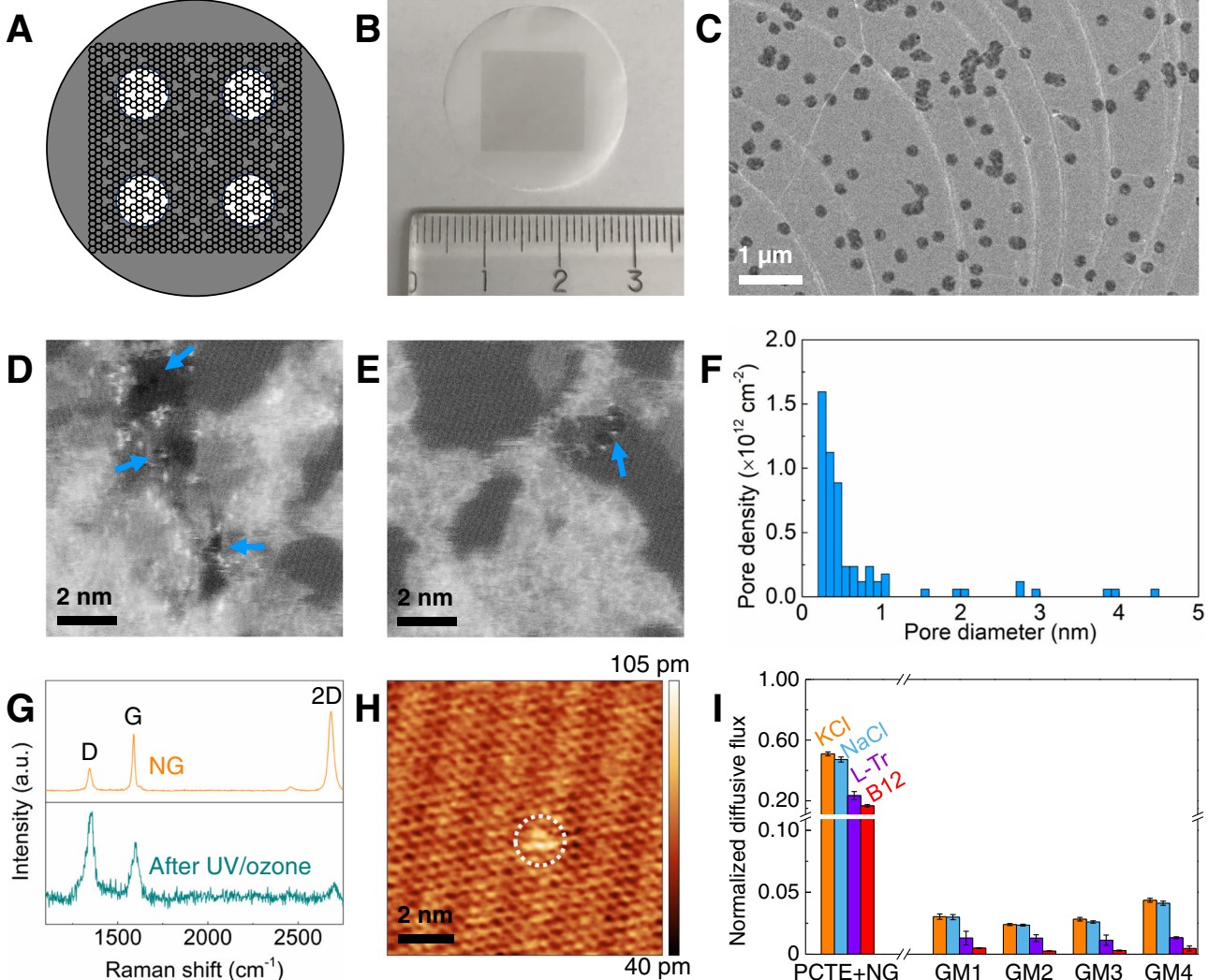

**Fig. 1 | Centimeter-scale, atomically thin graphene membrane with angstrom-scale pores. A** Schematic and **B** optical image of graphene membrane supported on polycarbonate track-etched (PCTE) supports. The black square in the image is graphene. **C** SEM image of graphene transferred onto PCTE support. The dark circles indicate PCTE cylindrical pores (~200 nm diameter) covered by graphene. **D**, **E** STEM images of graphene lattice after UV/ozone treatment for 25 min. Blue arrows indicate nanopores in the lattice. **F** Measured nanopore size distribution from STEM images. **G** Raman spectra of as-synthesized nanoporous CVD graphene (NG) and NG after 25 min of UV/ozone etching. An increase in the D peak indicates the introduction of additional defects in the graphene lattice. **H** STM image of NG on Cu foil after 25-min UV/ozone etching. The dotted, white circle indicates a vacancy defect. **I** Solute transport through the synthesized graphene membranes normalized with respect to bare PCTE supports: as-synthesized NG on PCTE support (PCTE + NG), and graphene membranes after UV/ozone and defect sealing via interfacial polymerization (GM1, GM2, GM3, and GM4). Diffusion-driven fluxes were measured for KCl (~0.66 nm), NaCl (~0.716 nm), L-Tr (~0.7–0.9 nm), and B12 (~1–1.5 nm). Error bars indicate one standard deviation. Source data are provided as Source Data files.

breathability in a protective garment application (1500–2000 g m$^{-2}$ day$^{-1}$). Hence, GMs offer the possibility of greatly enhanced protection against biological and possibly large chemical agents by size sieving (>0.66 nm) while simultaneously achieving higher WVTRs with enhanced thermal comfort compared to most state-of-the-art commercial breathable and protective materials.

Here, we note that the phase of water molecules transporting through the GMs is water vapor, because (i) the dew point under the experimental conditions used ($\overline{RH}$ = 30 or 40%, 30 °C, and atmospheric pressure) is ~10–15 °C, (ii) the pressure in the confined aperture is unlikely to reach the GPa range which is required to transform water vapor to liquid water at these conditions[9], (iii) prewetting was required to achieve liquid water permeation, thus indicating that the membrane pores are hydrophobic, and (iv) our molecular dynamics simulations for water vapor transport through graphene nanopores (with carbon, hydrogen, and hydroxyl terminal groups) in size range up to 10 Å show no water condensation despite the simulated higher relative humidity

of ~55% to promote condensation (see details in Supplementary Note 2). Finally, a modest increase in WVTR observed with increasing $\overline{RH}$ % for PCTE as well as GMs (Fig. 2B) indicates a transport mechanism inconsistent with water evaporation, since evaporative flux typically decreases with increasing RH (a trend that is opposite to the experimental observation).

After water vapor transport measurements (see Supplementary Fig. 3), we mounted each GM into a customized flow cell (Fig. 2D) to probe liquid-phase water transport. Initially, liquid-phase water transport across GMs was measured via forward osmosis (FO) with glycerol ethoxylate as the draw solution (see Methods section). The rationale for using glycerol ethoxylate as the draw solution is its relatively large average molecular weight (~1000, close to B12) and molecular diameter (~1.2 nm, close to B12), leading to negligible transport of the draw solute to the opposite side (feed side), to ensure accurate transport measurement (see Fig. 1I). Liquid water flux showed a linear increase with increasing osmotic pressure (4–26 bar, Fig. 2E) with water permeance

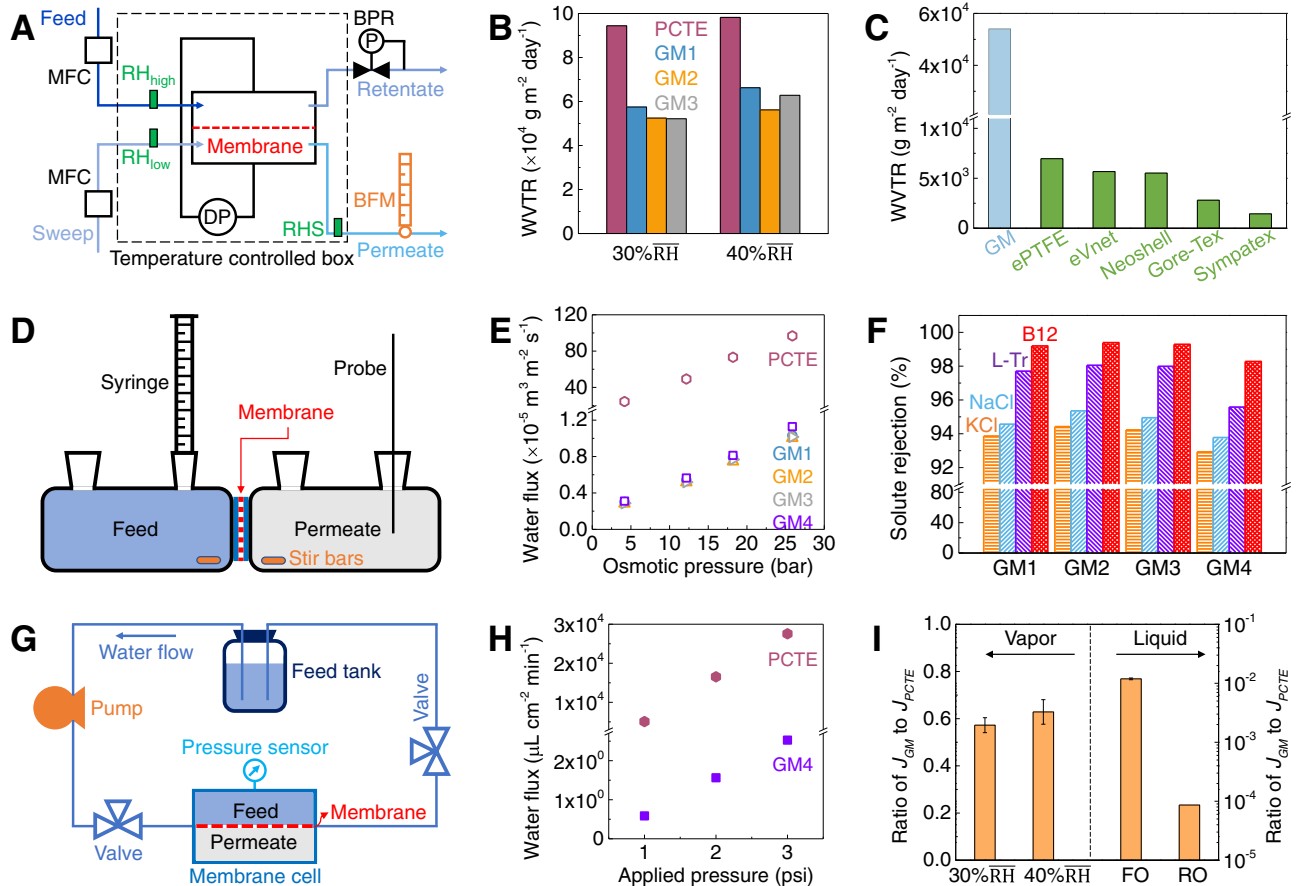

**Fig. 2 | Water vapor and liquid water transport through atomically thin graphene membrane with angstrom-scale pores. A** Schematic of the setup to measure water vapor transport−dynamic moisture permeation cell (DMPC) with mass flow controllers (MFCs), differential pressure meter (DP), back-pressure regulator (BPR), bubble flow meter (BPM), and relative humidity sensor (RHS). **B** Water vapor transmission rates (WVTRs) through bare PCTE and graphene membranes (GM) based on the effective membrane area (9.4% of total area) under different mean relative humidities (RHs) of 30 and 40%. **C** WVTRs of the fabricated GM (accounting for ~9.4% porosity of PCTE supports) and commercial breathable materials[54] (ePTFE, eVent, Neoshell, Gore-Tex, and Sympatex) measured under 30% $\overline{RH}$ with a constant RH difference of 50% across the membrane at 30 °C. Also, see Supplementary Fig. 2. **D** Schematic of the Side-Bi-Side diffusion cell used to measure liquid water as well as solute transport. **E** Water flux measured through bare PCTE and graphene membranes (GMs) under forward osmosis (FO) based on effective membrane area (9.4% of total area). Also, see Supplementary Fig. 5. **F** Solute rejections of KCl (-0.66 nm), NaCl (-0.716 nm), L-Tr (-0.7−0.9 nm), and B12 (-1−1.5 nm) through the fabricated GMs. Also, see Supplementary Fig. 7. **G** Schematic of the custom-made hydrostatic pressure-driven cross-flow reverse osmosis (RO) system for liquid water transport measurement. **H** Water flux measured through bare PCTE and graphene membrane (GM) under reverse osmosis (RO) based on effective membrane area (9.4% of total area). Also, see Supplementary Figs. 4 and 5. **I** Flux ratio of GM/PCTE for water vapor and liquid water. The GMs show significant water vapor flux (-57.3−62.9% of PCTE support) but very low liquid water flux (-1.2% of PCTE support under FO and -0.0086% of PCTE support under RO). All error bars indicate one standard deviation. Source data are provided as Source Data file.

~0.6 g m$^{-2}$ day$^{-1}$ Pa$^{-1}$ (Fig. 3B) and was found to be remarkably consistent across four different GMs. Considering the van der Waals diameter of water molecule ~0.28 nm[38], liquid water transport through GMs could arise from (i) selective water transport through small nanopores <0.66 nm, which block salt ions and organic molecules (see Figs. 1, 2), (ii) non-selective water transport through a very small number of large nanopores (>0.66 nm) which allow for leakage of salt ions (<4%) and organic molecules (<0.5−1%, see Fig. 1I), and (iii) negligible water transport through the POSS-polyamide (PA) plugs (see Supplementary Fig. 6) sealing large nanopores and tears in graphene.

To ensure no structural damage occurred to the GMs between water vapor and liquid water transport measurements, we also probed the GMs ability to reject model salts (KCl and NaCl) and organic molecules (L-Tr and B12) via FO in the same cell without unmounting the membrane (see Methods section). Indeed, the fabricated GMs show ~94% rejection of KCl, ~94.7% rejection of NaCl, ~97.3% rejection of L-Tr, and ~99% rejection of B12 (Fig. 2F). The negligible leakage of B12 (-1−1.5 nm) confirms that most pores in GMs (even after FO and water vapor transport) are sub-nanometer and even ions -0.66 nm face

significant transport resistance further confirming the potential of the fabricated GMs for breathable and protective applications.

Interestingly, when compared to the PCTE supports with ~200 nm pores, the GMs show a relatively mild reduction in water vapor transport (-57.3 and ~62.9% of PCTE support under 30 and 40% $\overline{RH}$, respectively, see Fig. 2B, I) but a significant drop in liquid water transport under FO (-1.2% of PCTE support see Fig. 2E, I). We note that, while leakage of glycerol ethoxylate (molecular diameter -1.2 nm) through GMs (with pores mainly <1 nm) is negligible (Fig. 2F and S8) in FO, glycerol ethoxylate can diffuse through the large pores (-200 nm) of the PCTE support into the waterside, thereby reducing the net osmotic driving force and leading to an underestimate of liquid water permeance across PCTE supports.

To rule out the effect from draw solution diffusion and obtain a more accurate value for liquid water permeance through PCTE support, we performed mechanical pressure-driven reverse osmosis (RO, see Fig. 2G) and obtained liquid water permeance -23,482 g m$^{-2}$ day$^{-1}$ Pa$^{-1}$ (Fig. 2H), consistent with the manufacturer's specification of -22,000 g m$^{-2}$ day$^{-1}$ Pa$^{-1}$ [58]. Notably, liquid water permeance across GM

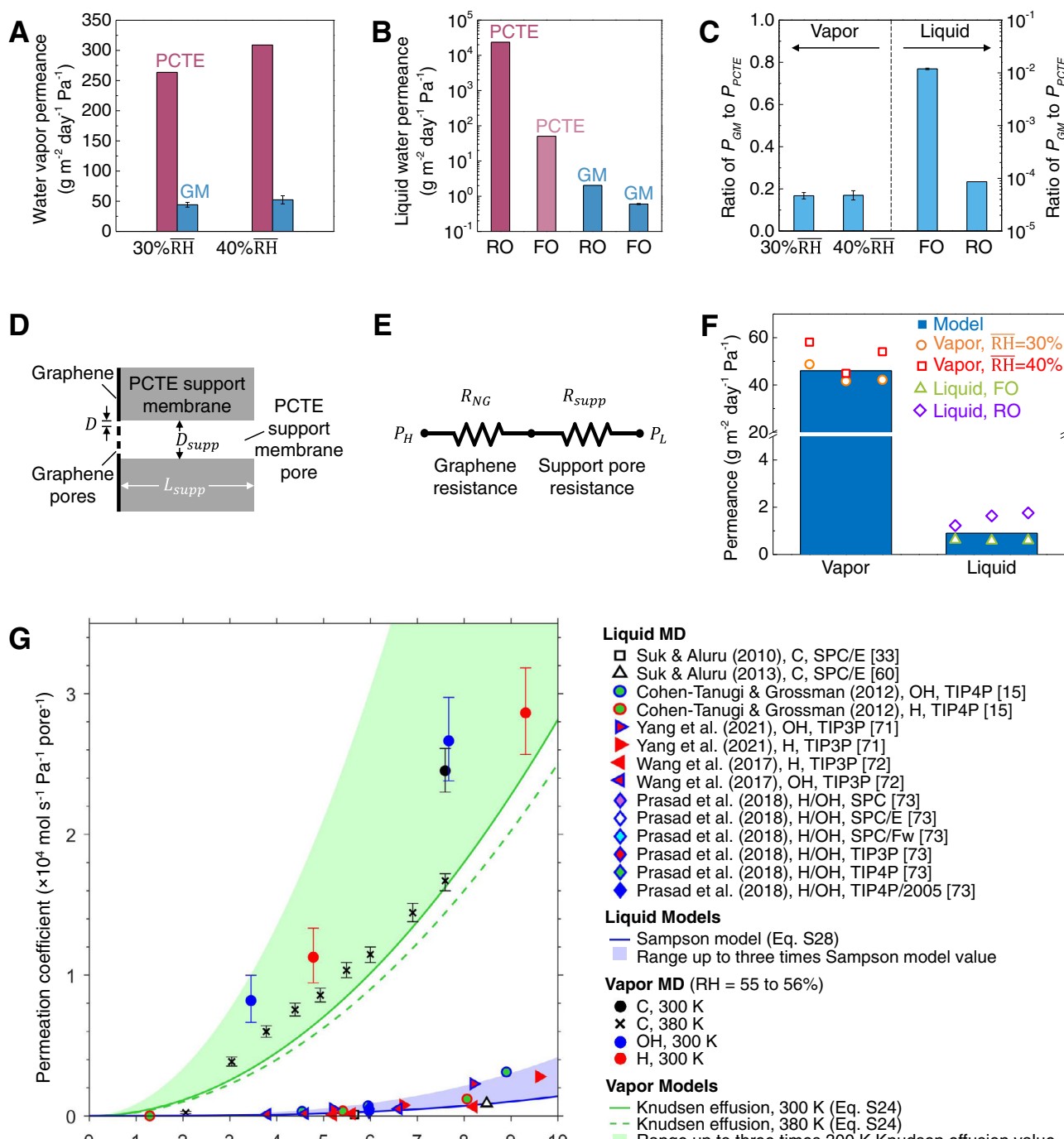

**Fig. 3 | Comparison of simulation models to measurements. A** Comparison of water vapor permeances between PCTE support and fabricated GMs under different mean relative humidity (RH) of 30 and 40%. **B** Comparison of liquid water permeances between PCTE support and fabricated GMs under forward osmosis (FO) and reverse osmosis (RO). **C** Permeance ratio of GM/PCTE for water vapor (after subtracting boundary layer resistance) and liquid water. The GMs show very high water vapor permeance (~16.8–17% of PCTE support) but very limited water permeance (~1.2% of PCTE support under FO and ~0.0086% of PCTE support under RO). All error bars indicate one standard deviation. **D** Structure of a single support PCTE membrane pore with graphene suspended over it. **E** Transport resistance

model ($P_H$ and $P_L$ are the FO or RO pressures, or partial pressures, on either side of the membrane). **F** Comparison of the transport model with the measured liquid and vapor permeances. Also, see Supplementary Fig. 12. **G** Comparison of liquid water[15,33,60,71–73] and water vapor permeation coefficients from molecular dynamics simulations (see Supplementary Note 2). Uncertainty bars for water vapor simulations show 95% confidence interval for a Poisson process. The legend format for liquid MD markers is "authors (year), pore terminal groups, water model [citation number]". Legend format for vapor MD markers is "pore terminal groups, temperature." Source data are provided as a Source Data file.

measured via RO was ~2 g m⁻² day⁻¹ Pa⁻¹ (as low as ~0.0086% of PCTE support, Fig. 2H, I) representing ~3.3× increase when compared to FO (~0.6 g m⁻² day⁻¹ Pa⁻¹), in broad agreement with prior reports of ~4.9× increase[31].

We also consider contributions from boundary layer resistance (relevant to the driving forces) during water vapor and liquid water transport through GM to understand the origin of the differences in transport rates. For water vapor transport, there is an air-side

boundary layer resistance existing at the membrane surfaces[54,59], which is membrane-independent and -104.5 s m$^{-1}$ for the DMPC system (see Supplementary information and Supplementary Fig. 9)[54]. After subtracting the boundary layer resistance from the total resistance, the water vapor permeance for GMs is -44.2 and -52.4 g m$^{-2}$ day$^{-1}$ Pa$^{-1}$ for 30 and 40% $\overline{RH}$, respectively, compared to PCTE -263.6 and -308.9 g m$^{-2}$ day$^{-1}$ Pa$^{-1}$ for 30 and 40% $\overline{RH}$, respectively (Fig. 3A). In the case of liquid water transport, a boundary layer resistance related to the variation of osmotic agent (draw solution) concentration at the GM surfaces is minimized by vigorous stirring (see Methods section, Supplementary Figs. 8, 10). Interestingly, the GMs show much lower liquid water permeance (0.6 g m$^{-2}$ day$^{-1}$ Pa$^{-1}$ under FO and 2 g m$^{-2}$ day$^{-1}$ Pa$^{-1}$ under RO in Fig. 3B) compared to PCTE support (50.5 g m$^{-2}$ day$^{-1}$ Pa$^{-1}$ under FO and 23,482 g m$^{-2}$ day$^{-1}$ Pa$^{-1}$ under RO in Fig. 3B). Taken together the permeance ratio for water vapor (GM/PCTE)$_{vapor}$ is -0.17 while that for liquid water using values from FO (GM/PCTE)$_{liquid, FO}$ is -0.012 and liquid water using values from RO (GM/PCTE)$_{liquid, RO}$ is -8.6 × 10$^{-5}$ (Fig. 3C).

We explore these differences in liquid water and water vapor permeance using a transport resistance model. Covering PCTE membrane support pores with nanoporous graphene reduces liquid water permeance to -1.2% (FO) or -0.0086% (RO) of the value without graphene, but water vapor permeance through the same membrane are only reduced to -16–17% (Fig. 3C), i.e., the same graphene pores present a more significant barrier to liquid water transport compared to water vapor transport in this system. The reason can be understood by considering the equivalent transport resistance network for this membrane.

Figure 3D shows the flow geometry through a single support pore in the PCTE membrane. To permeate the membrane, water molecules must pass through one of the nanopores in the graphene layer over the PCTE support pore and then through the support pore itself. The transport resistance to pass through the graphene acts in series with the PCTE support pore, as shown in Fig. 3E. In the equivalent electrical circuit model, mass flow rate ($\dot{m}$) replaces current, and pressure difference ($\triangle P$) replaces voltage difference such that $\dot{m} = \triangle P/R$, where $R$ is the flow rate resistance. The total mass flow rate through nanoporous graphene in series with a PCTE support pore is thus,

$$\dot{m}_{NG+supp} = \frac{\triangle P}{R_{supp} + R_{NG}} \qquad (1)$$

where $R_{supp}$ and $R_{NG}$ are the flow rate resistances for a PCTE support membrane pore and the nanoporous graphene over a support pore, respectively. Liquid and vapor flow through graphene nanopores occurs in different transport regimes, resulting in significantly different transport resistances that account for the difference in flow rates.

It is important to note that the graphene over each PCTE support pore will have several graphene nanopores of different sizes. We measured the pore size distribution by STEM imaging 89 graphene nanopores. After accounting for the difference in observed carbon atom diameter in the STEM and the van der Waals diameter (Supplementary Fig. 11), further reducing the pore diameter by the mean van der Waals diameter of water molecules, and excluding pores larger than 0.8 nm expecting them to be sealed by POSS, we obtained a density of water permeable pores of $n = 0.95 \times 10^{12}$ pores/cm$^2$ and average effective pore size for water permeable pores of $\langle D \rangle = 0.23$ nm. The average squared effective diameter and average cubed effective diameter are $\langle D^2 \rangle = 0.071$ nm$^2$ and $\langle D^3 \rangle = 0.026$ nm$^3$, respectively.

The total average flow rate through the graphene area over a PCTE membrane pore is then found by summing over this distribution,

$$\dot{m}_{NG} = nA_{supp} \frac{\sum \dot{m}_{pore}}{N_{pore}} \qquad (2)$$

where $\dot{m}_{pore}$ is the mass flow rate through a single graphene nanopore, $A_{supp} = \pi D_{supp}^2/4$ is the cross-sectional area of a PCTE support pore, $N_{pore}$ is the number of water permeable pores, and the summation here and throughout is over all water permeable pores.

Liquid water flow through the support PCTE membrane pores and graphene nanopores is reasonably modeled as continuum transport. The graphene nanopores are small enough that deviations from continuum theory are expected, as quantified by Suk and Aluru[60] using molecular dynamics simulations. However, for the composite membranes measured here, these differences have a much smaller impact on $\dot{m}_{NG+supp}$ than the difference between liquid and vapor flow so are omitted to simplify the discussion. Liquid flow in the cylindrical support pores can be approximated as Poiseuille flow (laminar pipe flow), giving an equivalent resistance of [61],

$$R_{supp} = \frac{32\mu L_{supp}}{\rho D_{supp}^2 A_{supp}} \qquad (3)$$

Flow through the graphene nanopores can be estimated from Sampson's expression for pressure-driven creeping flow through an infinitesimal thickness orifice plate[62],

$$\dot{m}_{pore} = \frac{\triangle P D^3 \rho}{24\mu} \qquad (4)$$

where $D$ is the graphene nanopore diameter. Substituting Eq. (4) into Eq. (2) leads to an equivalent nanoporous graphene resistance of,

$$R_{NG} = \frac{24\mu}{n\rho A_{supp} \langle D^3 \rangle} \qquad (5)$$

Substituting into Eq. (1), the liquid water permeance is given by,

$$\frac{\dot{m}_{NG+supp}}{\triangle P A_{supp}} = \frac{1}{\frac{\mu}{\rho} \left( 32 \frac{L_{supp}}{D_{supp}^2} + \frac{24}{n \langle D^3 \rangle} \right)} \qquad (6)$$

In water vapor transport, the osmotic pressure difference is replaced by the partial pressure (vapor concentration) difference, which drives the flow. Water vapor flows through the membrane pores near the free molecular regime. The support pore resistance is estimated from the equation for Knudsen diffusion[63],

$$R_{supp} = \frac{8}{3} \frac{L_{supp}}{A_{supp} D_{supp}} \sqrt{\frac{2\pi k_B T}{m}} \qquad (7)$$

The mass flow rate through a single graphene nanopore is modeled as Knudsen effusion (originated from Graham effusion[64]),

$$\dot{m}_{pore} = \triangle P \frac{\pi}{4} D^2 \sqrt{\frac{m}{2\pi k_B T}} \qquad (8)$$

Substituting Eq. (8) into Eq. (2) results in a nanoporous graphene resistance of,

$$R_{NG} = \frac{4}{\pi} \frac{1}{nA_{supp} \langle D^2 \rangle} \sqrt{\frac{2\pi k_B T}{m}} \qquad (9)$$

For water vapor flow, the flow rate expression in Eq. (1) simplifies to permeance of,

$$\frac{\dot{m}_{NG+supp}}{\triangle P A_{supp}} = \frac{\sqrt{\frac{m}{2\pi k_B T}}}{\frac{8}{3} \frac{L_{supp}}{D_{supp}} + \frac{4}{\pi} \frac{1}{n \langle D^2 \rangle}} \qquad (10)$$

As seen by comparing Eqs. (6) and (10), the permeance scales differently with pore diameter for liquid and vapor transport. Using the values of $D_{supp} = 0.2\,\mu m$ and $L_{supp} = 10\,\mu m$ for the PCTE membranes in this study, we obtain permeance values of 46 g m⁻² day⁻¹ Pa⁻¹ for water vapor and 0.9 g m⁻² day⁻¹ Pa⁻¹ for liquid water. These values are in reasonable agreement with the measured values of ~48 g m⁻² day⁻¹ Pa⁻¹ for water vapor and ~0.6 g m⁻² day⁻¹ Pa⁻¹ (FO) and ~2 g m⁻² day⁻¹ Pa⁻¹ (RO) for liquid water (Fig. 3F). This simple modeling reveals the reason for the significantly different flow rates for water vapor and liquid water transport through the graphene membrane. Liquid water flow occurs near the continuum regime, whereas water vapor undergoes nearly free molecular flow. For the membrane structure measured here, the resistance to water flow through the nanoporous graphene is ~80-fold higher than the resistance to vapor flow; the nanoporous graphene provides a much greater impediment to liquid flow than gas flow. The models in Eqs. (6) and (10) accounts for the measured permeance of liquid water and water vapor, as well as the large difference between the two. The significantly different flow rate scaling for liquids compared to low-pressure vapors means that the permeance through nanoporous graphene membranes will depend strongly on the fluid state.

To capture, in a simple way, the physical phenomena responsible for differences in flow rate reduction between liquid and vapor phase transport, a number of approximations have been made that will contribute to modeling errors. Notably, we have neglected deviations from continuum transport for liquid water flow through graphene nanopores and approximated vapor transport in the PCTE membrane pores as free molecular flow. Simulations have revealed a variety of nanoscale transport phenomena that occur during flow through graphene pores, including velocity slip, viscosity changes under nanoconfinement, the formation of dense liquid layers along solid boundaries, adsorption of gas molecules on graphene, surface diffusion, and interactions between fluid molecules and terminal groups on the edges of the pore[15,33,60,65–70]. The importance of such effects on transport rates was quantified by performing molecular dynamics simulations of water vapor permeation through various graphene nanopores (Supplementary Figs. 13–16 and Supplementary Tables 2, 3) and by compiling published simulation data for liquid water transport through graphene nanopores (Supplementary Fig. 17)[15,33,60,71–73]. In both cases, flow rate enhancements of up to a factor of ~3 compared to the simplified Knudsen effusion and Sampson flow models were calculated for some pores (Fig. 3G; details in Supplementary Note 2). Although these nanoscale transport phenomena influence the precise flow rates through graphene nanopores[74], the resulting factors of up to ~3 enhancements in both vapor and liquid flow rates are significantly smaller than the factor of ~80 difference measured between vapor and liquid transport rates. This order of magnitude difference is accounted for by the difference between the free molecular and near-continuum flow regimes in which vapor and liquid water molecules pass through the pores. This difference can be appreciated more simply by considering the analytical Knudsen effusion and Sampson flow models, although the molecular scale details would be necessary for more precise modeling.

Further sources of modeling error include uncertainty in the measured pore size distribution due to the limited sample size of graphene permeable pores that could be imaged and uncertainty in the precise nanopore size cut-off at which POSS will plug the pore. This uncertainty may mask flow rate enhancements compared to our simple modeling if they occur. Due to this uncertainty, the relative difference in liquid and vapor transport rates, and the overall order of magnitude of permeance, is more meaningful than the precise values calculated in Fig. 3F. The small inherent permeance of the POSS has also been neglected. Nevertheless, these simple models quantitatively explain the large difference in permeance measured for water vapor and liquid water transport based on the measured pore size distribution. We note that while we have included the resistance of the support membrane ($R_{supp}$) in the modeling, omitting it, in this case, would only change flow rate predictions by ~11% for water vapor and <0.1% for liquid water. Further details on these transport models are provided in Supplementary Note 1.

In conclusion, we report on orders of magnitude (~80×) differences in transport rates of vapor (~44.2–52.4 g m⁻² day⁻¹ Pa⁻¹) and liquid water (0.6–2 g m⁻² day⁻¹ Pa⁻¹) through angstrom-scale pores ~2.8–6.6 Å in centimeter-scale monolayer graphene membranes. Specifically, the permeance ratio for water vapor through the graphene membranes when compared to PCTE with ~200 nm pores (GM/PCTE)$_{vapor}$ is ~0.17, while those for liquid water using values from FO and RO are (GM/PCTE)$_{liquid,\ FO}$ ~0.012 and (GM/PCTE)$_{liquid,\ RO}$ ~$8.6 \times 10^{-5}$, respectively. Using a flow resistance model, we attribute the origin of these differences to distinct flow regimes during permeation, i.e., liquid water permeation occurs near the continuum regime while water vapor transport occurs in the free molecular flow regime. Finally, we leverage these insights to demonstrate centimeter-scale graphene membranes that can effectively block sub-nanometer (>0.66 nm) model molecules while permitting a water vapor transport rate of ~5.4–$6.1 \times 10^4$ g m⁻² day⁻¹, which is ~35 times higher than the generally accepted US military guideline for water vapor breathability in a protective garment (1500–2000 g m⁻² day⁻¹)[56] and an order of magnitude higher than most commercial breathable materials (1440–6900 g m⁻² day⁻¹)[54].

## Methods

### Graphene growth

Nanoporous graphene was synthesized on Cu foils at 900 °C using low-pressure chemical vapor deposition (LPCVD) as reported in detail elsewhere[32,42,43,51,75–80]. First, the Cu foil (99.9% purity, 18 μm thick, JX Holding HA) was pre-cleaned in diluted nitric acid (20%) via sonication for 4 min to remove surface oxides and contaminants, followed by rinsing in deionized (DI) water for 2 min and drying in air[32,42,43,51,75,76]. Next, the Cu foil was annealed in a 1-inch hot-walled tube furnace at 1060 °C for 30 min under 100 sccm H₂, and then cooled down to 900 °C (growth temperature). The graphene was grown under 3.5 sccm CH₄ and 60 sccm H₂ for 30 min, followed by another 30 min of growth with 7 sccm CH₄ and 60 sccm H₂. Finally, the foil was quench-cooled in the same growth atmosphere.

### Graphene transfer onto PCTE

Graphene transfer onto PCTE supports was performed via isopropanol-assisted hot lamination method[43]. Initially, graphene on the bottom side of Cu foil was removed by pre-etching the foil in 0.1 M of ammonium persulfate (APS) solution for 30 min, followed by rinsing the foil in DI water (two times, 10 min per time) and drying it in air[32,42,43,51,75,76]. Next, PCTE (~200 nm cylindrical pores, ~9.4% porosity, 10 μm thick, free of PVP coating, hydrophobic, Sterlitech Inc.) was placed against the graphene/Cu foil (graphene side facing up) and sandwiched between two pieces of weighing paper to build a paper/PCTE/graphene/Cu/paper stack. A small volume (50 μL) of isopropanol (IPA) solvent was added to the PCTE/graphene interface as the liquid heat transfer medium. The stack was laminated with Teflon protective layers at 135 °C using a TruLam TL-320E roll-to-roll office laminator. After peeling off the weighing paper, the Cu foil was fully etched by floating the PCTE/graphene/Cu stack on the APS solution. Finally, the PCTE/graphene stack was rinsed with DI water, followed by washing in ethanol and drying in air.

### Graphene transfer onto SiO₂/Si wafer for Raman spectroscopy

Graphene transfer onto SiO₂/Si wafers was performed using polymer assisted transfer method[32,42,43,51,75,76]. First, graphene on the bottom side of the Cu foil was removed as described above[32,42,43,51,75,76]. Polymethyl methacrylate (PMMA) in anisole (2 wt%) was drop-casted onto the graphene side of the pre-etched Cu foil, followed by drying in air. The

foil was subsequently etched in APS solution, and the obtained PMMA/graphene stack was rinsed in DI water for 10 min. Finally, the stack was transferred onto a SiO₂ (300 nm)/Si wafer, followed by baking in air, washing in acetone, and cleaning in IPA.

## Graphene transfer onto TEM grids

Graphene transfer onto TEM grids was carried out using the method reported elsewhere with some modifications[81–85]. Initially, CVD graphene on Cu foil was pre-etched to remove graphene on the bottom side as described above[32,42,43,51,75,76]. Next, the TEM grid (Ted Pella Inc. 658-200-AU with 1.2 μm holes) was placed against the graphene side of the pre-etched Cu foil with the Quantifoil carbon film contacting graphene. IPA (10 μL) was then added onto the stack to wet the graphene-grid interface. The stack was allowed to dry for 2 h at room temperature and then annealed at 80 °C for 30 min to enhance the adhesion between the graphene and the grid. Finally, the Cu foil was fully etched in APS solution, rinsed thoroughly in two subsequent DI water baths, followed by rinsing with IPA and drying in air.

## UV/ozone treatment

UV/ozone etching was performed in a UV/ozone cleaner (Jelight Model 30) for 25 min to introduce new defects and enlarge existing nanopores in the graphene lattice[32].

## Interfacial polymerization

Interfacial polymerization (IP) was performed based on the methods previously described elsewhere[32,42,43,50,51,75,76,86]. PCTE/graphene membrane after UV/ozone treatment was initially annealed at 105 °C for 12 h. IP reaction was carried out in a Franz cell (PermeGear, Inc.) with octa ammonium polyhedral oligomeric silsesquioxane (POSS, Hybrid Plastics, AM0285, 0.4 g) in water (20 mL, pH 10.7 with the addition of NaOH) as the aqueous phase and trimesoyl chloride (TMC, Alfa Aesar, 4422-95-1, 0.035 g) in hexane (10 mL) as the organic phase.

## Characterization

SEM images of graphene on PCTE supports were recorded by using a Zeiss Merlin Scanning Electron Microscope with a Gemini II Column operated at 1–2 kV.

Raman spectra were acquired using a Thermo Scientific DXR Confocal Raman spectrometer with a 532 nm laser source.

STEM images were collected by using a Nion UltraSTEM 100 aberration-corrected scanning transmission electron microscope (STEM), operated at 60 kV in the Center for Nanophase Materials Sciences at Oak Ridge National Laboratory[32,42,51,75,76]. The graphene samples on TEM grids were annealed overnight under vacuum at 160 °C before imaging[32,42,51,75,76]. The pore size of each nanopore in collected STEM images was estimated[15] by converting the manually-measured open area (A) into an effective diameter via $d_{pore} = \sqrt{4A/\pi}$ (Fig. 1F). We also calculated the pore size (Supplementary Fig. 1) by adding the carbon electron diameter (0.13 nm) and subtracting carbon van der Waals diameter (0.34 nm)[32,38,84,87]. The pore density was computed by dividing the total number of imaged pores by the total effective area (not covered by contaminants) of the acquired images.

STM images were obtained with an Omicron variable temperature scanning tunneling microscope (VT-STM) at room temperature in the Center for Nanophase Materials Sciences at Oak Ridge National Laboratory[32,42]. The samples were annealed under vacuum at 420 °C for 3 h before imaging.

## Water vapor transmission rate measurements

Water vapor permeation through the as-fabricated graphene membranes (PCTE+NG+UV/ozone 25 min + IP membranes, named here as GMs, Supplementary Fig. 3) and PCTE control was measured with a cross-flow Dynamic Moisture Permeation Cell enclosed in a thermostated box (Fig. 2A)[52], as described previously[54,59]. Both sides of the membrane were exposed to 1000 sccm (Q) N₂ gas streams, the relative humidity (RH) of which was controlled by mixing a wet gas stream with a dry gas stream in the desired proportions[54]. During all tests, a 50% RH difference was maintained across the membrane, and the temperature was kept at 30 °C. To ensure a purely diffusive water vapor transport, the pressure gradient across the membrane was set to zero with a back-pressure regulator and measured with differential pressure transmitters (Omega PX409-001DWUI). RH of the gas stream entering the high-humidity cell side was monitored with an Omega RH-USB humidity sensor, while RHs of the gas stream flowing through the low-humidity side were monitored by two Vaisala HM70 RH sensors before and after the cell. Once the desired flow rates and relative humidity were established, the system was left to stabilize for >30 min before recording the water concentration readings from the two Vaisala RH sensors. The difference between these two water concentrations (ΔC) was used to calculate the mass flow rate ($\dot{m}$) of water vapor diffusing across the membrane, $\dot{m} = Q \times \triangle C$. The measured water vapor transmission rates were based on the entire membrane area, without adjusting for differences in porosity among the membrane types; while the normalized water vapor transmission rates were based on the effective membrane area (accounting for ~9.4% porosity of PCTE supports). Measurements were performed at two mean RHs (30 and 40%), which was defined as the average of the two incoming gas stream RHs.

## Pressure and diffusion-driven solute transport measurements

After water vapor transport measurements, we carefully unloaded GMs (Supplementary Fig. 3), cut off the backup material and epoxy area, and then mounted each GM onto the customized diffusion cell system (Fig. 2D) for subsequent solute diffusion, liquid water transport, and solute rejection measurements. The GM region for water vapor transport measurement is a large square with each side ~0.9 cm (Supplementary Fig. 3), leaving sufficient GM area after trimming to cover the 0.5 cm orifice of the diffusion cell system, thereby allowing liquid-phase transport measurements across identical GMs used in water vapor transport measurements.

Pressure-driven ethanol transport and diffusion-driven solute transport measurements across the fabricated membranes were all performed as reported in detail previously[32,42,43,51,75,76,84,85,88,89]. A customized 7 mL Side-Bi-Side glass diffusion cell (5 mm orifice, Perme-Gear, Inc.) with a gastight syringe (250 μL, Hamilton 1725 Luer Tip) installed onto the short open port of the left cell (leak-free connection, sealed with epoxy) as shown in Fig. 2C was used for transport measurements. The membrane was installed between two diffusion cells (with the graphene side facing left), followed by clamping the cells in the diffusion system. The feed solution was always introduced into the left cell (graphene side) with magnetic Teflon-coated stir bars stirring vigorously at 1500 rpm in both cells to prevent concentration polarization (see Supplementary Fig. 10).

For pressure-driven ethanol transport measurement[43,76,89], pure ethanol (200 proof) was used to wash the system three times before measurement. Both cells were subsequently filled with pure ethanol (the ethanol level was 250 μL in the graduated syringe) and the ethanol height difference generated the hydrostatic pressure gradient. During the measurement, a digital camera was used to record the ethanol meniscus level decrease along the syringe every 1 min. The ethanol permeance was computed by $p = (\triangle V/\triangle P)/(\triangle t \times A_{effective})$, where $p$ is the ethanol permeance, $\Delta V$ is the ethanol volume change (decrease), $\Delta P$ is the hydrostatic pressure difference across the membrane, $\Delta t$ is the time interval (1 min), and $A_{effective}$ is the effective membrane area. The normalized flux was calculated by dividing the ethanol permeance of each membrane by the ethanol permeance of the PCTE substrate[43,76,88,89].

Prior to diffusion-driven solute transport measurements, the system was washed with DI water 5 times to completely replace

ethanol residue and wet the PCTE cylindrical pores. Four model solutes were specifically selected for measuring diffusion-driven transport: KCl (Fisher Chemical, 7447-40-7, salt, hydrated diameter of $K^+$ ~0.662 nm and $Cl^-$ ~0.664 nm)[38], NaCl (Fisher Chemical, 7647-14-5, salt, hydrated diameter of $Na^+$ ~0.716 nm and $Cl^-$ ~0.664 nm)[38], L-tryptophan (L-Tr, VWR, 73-22-3, amino acid, ~0.7–0.9 nm)[51], and Vitamin B12 (B12, Sigma-Aldrich, 68-19-9, vitamin, ~1–1.5 nm)[51]. For measuring salt (KCl or NaCl) transport[32,42,43,51,75,76,84,85,88], 7 mL of salt solution (0.5 mol $L^{-1}$ in DI water) was filled into the feed side and 7 mL of DI water was filled into the permeate side, with a conductivity meter probe (connected to a Mettler Toledo SevenCompact S230 conductivity benchtop meter) immersed in the permeate side to record the conductivity every 15 s for 15 min. For measuring organic molecule (L-Tr or B12) transport[32,42,43,51,75,76,84,85,88], 7 mL of organic molecule solution (1 mmol $L^{-1}$ in 0.5 mol $L^{-1}$ KCl) was filled into the feed side and 7 mL of KCl solution (0.5 mol $L^{-1}$) was filled into the permeate side, with a fiber optic dip probe (attached to an Agilent Cary 60 UV-vis Spectrophotometer) immersed in the permeate side to collect the absorbance spectra in the range of 190 to 1100 nm every 15 s for 40 min. Different UV-vis positions were used for measuring the intensity differences of corresponding species: 710 nm for DI water (reference wavelength)[32,42,43,51,75,76,84,85,88], 279 nm for L-Tr[32,42,51,75,76,85,88], and 360 nm for B12[32,42,43,51,75,76,85,88], respectively. The flow rate of each solute was computed via the slope of concentration change in the permeate side, while the normalized flux was calculated by dividing the slope of the fabricated membrane by that of the PCTE support membrane[32,42,43,51,75,76,84,85,88]. All the measurements were repeated in triplicates to obtain average values and standard deviations[32,42,43,51,75,76,84,85,88].

## Liquid water transport measurements

To reliably measure the transport of only liquid water, we wet the GMs fully with ethanol (a low surface tension solvent) and then rinsed thoroughly with DI water before liquid-phase transport measurements.

Liquid water transport experiments were performed using the same setup mentioned above (the customized 7 mL Side-Bi-Side glass diffusion cell shown in Fig. 2C) via forward osmosis with glycerol ethoxylate (Sigma-Aldrich, 31694-55-0, average molecular weight Mn ~1000) as the draw solution[32,84]. The feed side was filled with 8 mL of DI water, followed by sealing with a rubber plug (resulting in the rise of water level in the syringe); while the permeate side was filled with 8 mL of draw solution (10–30 wt% glycerol ethoxylate in DI water). The generated osmotic pressure difference (~4–26 bar) acts as a driving force to induce water to flow from the feed side through the membrane and into the permeate side. The water transport leads to the drop of water meniscus level along the syringe (feed side), which was recorded every 2 min (for GMs) via a digital camera. We note that glycerol ethoxylate could diffuse through the large pores (~200 nm) of PCTE support (non-selective to glycerol ethoxylate) into the waterside, leading to the decrease of an osmotic driving force, hence we recorded the water meniscus change every 2 s for bare PCTE support and calculated the water transport based on the data recorded in the initial 10 s (almost linear) to reduce the influence of glycerol ethoxylate diffusion on water transport.

The osmotic pressure was calculated by the following relation[84] $log\Delta\Pi = 4.87 + 0.8 \times (wt\%)^{0.34}$ where $\Delta\Pi$ is the osmotic pressure with the units of dyne/$cm^2$. Water flux was computed by the following equation[32,84] $j_{water} = \frac{\Delta V}{(A \times \gamma \times \Delta t)}$ where $\Delta V$ is the change of water volume along the graduated syringe, $A$ is the orifice area of the diffusion cell, $\gamma$ is the porosity of PCTE support (9.4%), and $\Delta t$ is the measurement time. Water permeance was calculated by dividing representative water flux by corresponding osmotic pressure[32,84].

Liquid water transport experiments were also carried out using a homemade hydrostatic pressure-driven cross-flow reverse osmosis (RO) system (see Fig. 2G). The membrane (PCTE and GM4) was supported by a Si mask (~50% porosity, providing mechanical support) and sandwiched between two polydimethylsiloxane (PDMS) gaskets to ensure the system is leakage-free. Before measuring liquid water transport, the mounted membrane was rinsed with ethanol (200 proof) and deionized water (Milli-Q) thoroughly. The feed water was circulated using a rotary pump at a rate of 20 ml $min^{-1}$ with different hydraulic pressures (1, 2, and 3 psi) applied on the feed side of the membrane. The liquid water permeance was obtained by dividing the water flow rate (measured by the meniscus level change along a graduated syringe on the permeate side) per hydraulic pressure by the effective membrane area (taking into account the ~9.4% porosity of PCTE). The diffusion-driven solute transport and the liquid water transport via FO across the GM4 membrane were performed after RO on the same membrane to confirm membrane integrity.

## Solute rejection measurements

Solute rejection experiments were also performed using the same setup (shown in Fig. 2C) via forward osmosis with glycerol ethoxylate (Sigma-Aldrich, 31694-55-0, average molecular weight Mn ~1000) as the draw solution[32,84]. For salt rejection experiments, the feed side was filled with 8 mL of salt solution (KCl or NaCl, 16.6 mM), followed by sealing with a rubber plug (resulting in the rise of solution level in the syringe); while the permeate side was filled with 7.8 mL of draw solution (25 wt% glycerol ethoxylate solution)[84]. The conductivity probe was immersed in the permeate side to measure the conductivity change every 15 s. For organic molecule rejection experiments, the feed side was filled with 8 mL of organic molecule solution (L-Tr or B12, 1.3 mM), followed by sealing with a rubber plug (resulting in the rise of solution level in the syringe); while the permeate side was filled with 7.8 mL of draw solution (25 wt% glycerol ethoxylate solution)[84]. A fiber optic dip probe (attached to an Agilent Cary 60 UV-vis Spectrophotometer) was immersed in the permeate side to record the absorbance spectrum change in the range of 190 to 1100 nm every 2 min[84].

The solute rejection was computed using the following equation[84], $S_{rejection} = \left(1 - \frac{j_{solute}/j_{water}}{C_f}\right) \times 100\%$, where $C_f$ is the initial solute concentration on the feed side, $j_{solute}$ and $j_{water}$ are solute flux and water flux, respectively (Fig. 2E)[84]. The solute rejection was also calculated by another equation[31], $S_{rejection} = \left(1 - \frac{C_p}{C_f}\right) \times 100\%$, where $C_p$ is the solute concentration on permeate side after 24 h, and $C_f$ is the initial solute concentration on the feed side (Supplementary Fig. 7).

## Data availability

All data are available in the manuscript, the supplementary materials and from the authors on request. Source data are provided as a Source Data file. Source data are provided with this paper.

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

## Acknowledgements

The use of Vanderbilt Institute of Nanoscale Science and Engineering CORE facilities and Prof. Carlos Silvera Batista's Lab for UV/ozone etching are acknowledged. This work was supported in part by NSF CAREER award #1944134, ACS PRF Grant number 59267-DNI10, and faculty start-up funds to P.R.K. from Vanderbilt University. The STEM and STM experiments were conducted at the Center for Nanophase Materials Sciences at Oak Ridge National Laboratory, which is a DOE Office of Science User Facility. Molecular dynamics simulations were performed in LAMMPS[90] (http://lammps.sandia.gov) and visualized in VMD (visual molecular dynamics)[91]. This work made use of computing resources of the Shared Hierarchical Academic Research Computing Network (SHARCNET: www.sharcnet.ca) and Compute Canada (www.computecanada.ca). F.F. and M.L.J. acknowledge financial support from the Chemical and Biological Technologies Department of the Defense Threat Reduction Agency (DTRA-CB) via grant BA12PHM123 in the "Dynamic Multifunctional Materials for a Second Skin D[MS]²" program. Work at LLNL was performed under the auspices of the US Department of Energy under contract DEAC52-07NA27344.

## Author contributions

P.R.K. conceived and supervised the project. P.C. fabricated graphene membranes, performed Raman and SEM characterizations, carried out the diffusion, forward osmosis and reverse osmosis transport measurements of liquid water, and analyzed the results. F.F and M.L.J measured the water vapor transport and wrote the corresponding sections. W.K. and A.-P.L. performed scanning tunneling microscopy measurements. J.C.I. performed scanning transmission electron microscopy measurements. M.S.H.B. performed the transport resistance model calculations, molecular dynamic simulations, and wrote the corresponding parts. All the authors were involved in the analysis and discussions of the results. P.C. and P.R.K. wrote the manuscript with input from all co-authors.

## Competing interests

P.R.K. acknowledges a stake in a company commercializing 2D materials. The remaining authors declare no competing interests.

## Additional information

**Correspondence and requests** for materials should be addressed to Piran R. Kidambi.

