## [Peer Review File · Nature Communications]

REVIEWER COMMENTS

Reviewer #1 (Remarks to the Author):

Authors reported that water vapour transport is orders of magnitude higher (x80) than the liquid water transport. These observations for water vapour transport already made by few previous studies i.e. by Celebi et al ($5 \times 10^6 \text{ g m}^{-2} \text{ day}^{-1}$ for bilayer graphene with 200nm pores, Ref.39) and liquid water transport by osmotic pressure by Yang et al ($5 \text{ g m}^{-2} \text{ day}^{-1} \text{ Pa}^{-1}$, ref31: and this value is in agreement with the value $0.6 \text{ g m}^{-2} \text{ day}^{-1} \text{ Pa}^{-1}$ reported in this draft by authors) through nanopores of monolayer graphene. From experimental point of view, report of water vapour transport through nanopores ($<0.66 \text{ nm}$) of single layer graphene is a new theme here. Then, authors invoked series resistance model to shed insights on the mechanisms of water and vapour transport through nanoscale confinement and they attributed to different transport regimes based on the higher flow rates of vapour than liquid water.

However, authors need to address the following comments to support their findings

1. Authors claim that pore sizes are <6.6 Angstroms. This could in principle mean there are 1 angstrom sized pores for instance, which is probably misleading. To be precise and clear, the authors could specify a range (max and min size of pores, probably 5 \AA to 6.6 \AA , as the polymerizations seems to close the pores bigger than 5 \AA). This change has to be made in abstract and in the main text.
2. Water condensation – in page 4, line150: The authors say “Here, we emphasize that the phase of water molecules transporting through the GMs is water vapour, because i) the dew point under the experimental conditions used (RH= 30% or 40%, $30 \text{ }^\circ\text{C}$, and atmospheric pressure) is $\sim 10\text{-}15 \text{ }^\circ\text{C}$ ”. With such condition of high moisture content, there is also a very likely possibility of pore condensation. Especially as the pores are much smaller (which authors points out in the very next sentence), the condensation can happen at much smaller humidity. Have the authors considered this possibility? How do they justify that there is no mixed conduction of water + vapour.
3. For water vapour transport through nanopores of $<6.6 \text{ \AA}$, water molecules will get absorbed inside the pore surface which will reduce the effective pore size by its dimension 3\AA (roughly), therefore the effective pore diameter (free space) left for water vapour molecules (WVMs) will be $<3.3\text{\AA}$ and within such a confinement regime it's reasonable to assume that WVMs will be bonded by hydrogen bonds and therefore molecules will not be free essentially, probably becomes bigger water droplet inside nanopore of such tight confinement regime with 30%-40% RH. It has been observed/known that for graphene/graphene derivatives, such small pore membranes could trap water molecules for

few days once it is exposed to water vapour. In such scenario, assumptions of free molecular transport probably might not a good starting point (using those corresponding equations).

4. As a continuation of the above discussions, its might be useful to show $1/\sqrt{T}$ dependency to prove its free molecular transport of water vapour through nanopore (by comparing Eqn6 and Eqn 10 in the main draft), as only the proof (rough estimations) comes from those equations provided by authors.

5. A few factors that could enhance the water vapour transport:

(a) Water evaporation on permeate side. It is possible that water evaporation flux happens from permeate side of the membrane for such tighter nanopores (in addition, the sweeping gas flows on the permeate side with lower RH) and thereby authors see much enhanced transport rate as compared to liquid water transport (evaporation driven enhanced flux). Authors need to address this point. A clear distinction is needed here to differentiate the water vapour molecular transport or water evaporation occurring on the other side of the membrane as transport rate of water evaporation will also give orders of magnitude difference than the rate for liquid water transport.

(b) Enhanced interaction with functionalization groups present at the pore mouth (on the feeding site) due to UV/ozone etching. Pores created with etching method contains functional groups (FGs) attached to its pore edges which will modify the water vapour transport. The interactions between FGs with the water molecules would exert additional forces which will, in-turn, render the transport rate/mechanisms. These forces will be negligible when bulk liquid water comes in contact with those attached FGs, because those interactions for water vapour molecules will change their other degrees of freedom (rotational and translational motions, polarization). Therefore, direct comparisons between transport rates for water vapour molecules and liquid water without counting these effects into calculations might not shed more insights onto the transport mechanisms through such tighter confinements.

(c) Also, surface diffusion could enhance the water vapour transport rates. Therefore, counting all these above mentioned factors towards the enhancement of water vapour transport, the actual resistance could be similar in range for liquid water and water vapour transport.

6. The liquid bulk water (on feeding site) has high dielectric constant (KDC) ~ 80 . Also, we know that when water molecules pass through tighter confinements it has very low KDC (around ~ 2) as previously reported. In general, the faster transport of water molecules occur through nanochannel because of reduced KDC (associated with ordered/layered water molecules inside nanopore). Therefore, the observed less transport rate for liquid water could be due to the presence of other limiting steps.

7. Use of large molecule, glycerol ethoxylate as a draw solution:

(a) In a way, this large molecule is blocking the membrane pores from the permeate side, which can hinder the water from feed to pass freely to the other side. What is the rationale behind using such large molecule? Is this mainly the reason for lower liquid water permeation, as the flow is being hindered? The authors should do a control experiment where they use smaller molecule on the permeate side, and compare the water flow.

(b) It is nice that authors have compared the experimental values of water and vapour transport, with those estimated from classical Poiseuille equation and Knudsen. However, in the case of liquid water transport, the role of slip (and surface diffusion if any) is not considered, which is expected to affect the liquid water flow estimates (enhanced flow rate is expected due to slip).

(c) For liquid water measurements (FO), on permeate side there is already draw solution. Now if we don't put any draw solution, then in principle liquid water transporting via GMs will exist the pore via evaporation flux which could show similar rate as water vapour experiments show. In such scenario, it's not easy to distinguish transport mechanisms of water molecules through/inside the nanopores.

8. In Fig. 2, the data for FO and moisture transport for various GM samples is "remarkably" consistent as authors also pointed out. How is this feasible? The support membranes themselves might have some pore distribution range, so each GM sample may have slightly different number of effective pores. Additionally the method of pore creation, interfacial polymerization (IP) will lead to further distribution in the pore densities (and sizes) in each sample. This is also evident in Figure 1, where the GM pore density is shown. How is it possible that four different samples show same flux, with such accuracy? Authors need to explain this, and give realistic distribution of effective pore density ranges per sample.

9. Limit of detection: The measurement techniques for salt rejection and organic molecule rejection are based on conductivity and optical probe. However, the authors should mention the limit of

detection/accuracy of the measurement techniques. If there is even a small amount of salt permeating through the pores, this might not be captured as it is diluting a larger volume of permeate solution. In the experimental section, the authors mentioned about normalization of concentration profiles of salts, however data should be presented for these control experiments. Also, the authors should present the limit of detection for their both conductivity and optical probe methods using impermeable samples (perhaps could be same graphene membranes without pores), and compare with their GM samples.

10. For the resistance model, the resistance for GMs is much larger than the PCTE support ($R_{NG} \gg R_{supp}$) for both the measurements, liquid water and water vapour transport measurements. The quantitative analysis could have been shown only by the resistance of GMs (R_{NG}) instead of invoking series-resistance model. Therefore, in order to justify the series resistance model (R_{supp} could have been removed) authors need to provide more information towards this.

11. It is likely that vapour transport is higher than liquid, but whether it is as high as 80x, needs to be proved more cautiously. Especially, considering the measurement of liquid water in FO could be lower due to the large permeate molecule blocking the pores on the membrane (along with comments-5&7).

Although, authors quantitatively showed the agreement between resistance model and their different transport regimes, the microscopic picture is still lacking here for water vapour transport through such tighter confinements. As mentioned in comment-1, such nanopores always trap water vapour molecules inside and stays longer once it is exposed to water vapour. Therefore the measurements after that (on exposed membrane) will not be free molecular diffusion transport. Also, there are other mechanisms (comment-3) which will complicate the water vapour transport.

Reviewer #2 (Remarks to the Author):

Description of transport dynamics in nanofluidics requires two variables largely: length and lateral span (e.g., diameter) of a nanochannel. When both are quite larger than molecules passing the nanochannel, transport dynamics can be described by continuum theories. For nanochannels that puts confinement of one of the variables, transport is known to show abnormal diffusion (e.g., carbon nanotube of tiny diameters) or ultimate permeation (e.g., supernanometric graphene orifice). What has not yet been well explored is the case when both variables are limited to the size commensurable with transmitting molecules own sizes, a transport regime where experiments and molecular simulations are both rare. In this manuscript, Kidambi and colleagues are using state-of-the-art meticulous methods to obtain transport data in the unknown transport regime. Instead of

varying the orifice size with producing obscure data, they took a simple but clear path of comparing transports of liquid and vapor for one (range of) pore size. In this way, the authors report that a subnanometric orifice of graphene allows transports of liquid water and water vapor (moisture) follows continuum theory and free molecular flow physics, respectively. They fabricated subnanometric graphene orifice samples according to their previous method, with confirming the orifice pore size around 0.66 nm via hindered diffusion of reference ions/salts. For liquid water, they employed forward osmosis and pressure-driven setups to confirm each other. For water vapor transmission rate, an elaborate experiment by use of dynamic moisture permeation cell. By comparing experimental data with basic transport theories, they add a data of liquid water transmission that agrees with a continuum hydrodynamic theory. The fact that vapor transmission follows effusion is not surprising, and yet how faster the vapor transmission is than the liquid water permeation for ca. 0.6-nm-wide graphene orifices might serve the research community as a useful data in the future.

After several corrections made to the manuscript, I recommend this manuscript for potential publication.

Suggestions for minor revision:

MVTR (moisture vapor transport rate) should be changed to WVTR (water vapor transmission rate).

“Knudsen effusion” should be replaced by “Graham effusion”.

Liquid water permeance data of Figure 3F should be shown in log scale for clarity, perhaps in the supplementary information.

Reference section needs polishing. Most of references from the Science magazine come with typos next to the journal name. Author names of ref. 64 contain typos.

Reviewer #3 (Remarks to the Author):

In this manuscript, the authors developed graphene membranes with pore size of less than 0.66 nm and transferred them to a polycarbonate support layer. The pore size larger than the specified dimension is repaired through size-selective interfacial polymerization. Transport of water in two states of vapor and liquid is measured through these pores. It has been shown that while liquid water transport through these pores is limited, the vapor flux is in the same order of other nanoporous materials. That is, these nanoporous membranes are selective in water vapor transport and exclude liquid water and larger molecules. These membranes could provide a new material

paradigm for a wide range of applications including desalination and protective fabrics. I support publication of this work after addressing the following comments:

- 1) The approximation of continuum fluid for 0.66 nm confined liquid does not seem correct. How do you define viscosity in these length scales? And what is the value? The authors need to conduct MD simulations for better understanding of liquid transport in these scales.
- 2) Is it only the pore dimension that matters here? Or the interaction of water molecules with walls plays a role? For example, could one achieve the same performance through Zeolite structures?
- 3) How does temperature affect the vapor transport through these pores? Does interaction with the walls affect the mass flux. In the free molecular transport model, the wall interactions are not taken into account.
- 4) How do you ensure perfect attachment of the Graphene membrane to the polycarbonate and no leaking through the imperfection between these two?

Reviewer #4 (Remarks to the Author):

Differences in water and vapor transport through Angstrom-scale pores in atomically thin membranes

P. Cheng et al.

Summary

The Authors report an arsenal of experimental results conducted for water transport through single layer graphene membranes, with controlled defects. The defects lead to rather small pores, which are small enough for preventing, e.g., salt ions transport. Significant characterisation is conducted for these materials, in order to interpret experimental data for transport. The results show significant differences between liquid vs gaseous transport. To enhance the interpretation of the experiments, macroscopic models are used, in which the pore sizes and membrane thickness are used as parameters. The models provide data in good agreement with the experiments, suggesting that the interpretation of the experimental data is reliable.

Recommendation

This is a very well written paper, with many relevant references. The Authors place the work in the context of recent contributions, and clearly identify some discrepancies in the literature. The results are frequently interpreted referencing to simulation results, also from the literature, and the presentation seems to be consistent. The subject matter is of high practical importance, and the experiments seem to be conducted convincingly.

Thus, I recommend publication of this piece of work once the Authors have addressed my comments below.

Details

The Authors seem to have made significant advancements along the lines of producing graphene membranes of macroscopic size. One 2016 Annual Review of Chemical and Biomolecular Engineering (Joly et al.) on the carbon-water interface suggested that this was a practical hurdle that needs to be overcome. Could the Authors comment on how large membranes they would be able to build?

The last sentence of the abstract, in my opinion, needs to specify what molecules larger than water have been blocked by the membranes being used here.

In the introduction, the Authors refer to relevant literature for liquid vs. vapour water transport through membranes. There is a recent work by Tuan Ho and colleagues at Sandia who investigated similar differences, although through a clay membrane (if memory serves me correctly). They also report large differences in transport rates depending on the experimental conditions. I think that is a good reference to contrast different transport mechanisms.

The graphene group in Manchester produces graphene membranes for a variety of applications. Am I correct to interpret that the main difference is that the membrane used here is a single layer graphene, while those in Manchester tend to be multiple graphene layers deposited on each other? Could the Authors comment on how the results presented would change, in their opinion, if the 'stack of graphene layers' was used instead of the single graphene membrane?

The Authors compare the high flow rate in their membrane to values obtained for commercial MVTR samples, which have larger pores. I think that for this comparison to be more convincing, the Authors should also report the surface density of the pores in the two substrates.

To study vapor transport, the Authors chose conditions of 30-40% relative humidity. There are a number of simulation studies conducted using the grand canonical Monte Carlo formalism (see, e.g., the group of Gubbins) that investigated water sorption in carbon pores of various geometries. Could

the Authors comment on whether at the conditions chosen here water confined in the graphene pores is expected to be high- or low- density, thus reminding of liquid vs vapor water?

The Authors confirm the size of the pores in graphene by measuring the transport of KCl ions. This reminded me of simulation results (Konatham et al., Langmuir, many years ago) where ions transport was simulated through graphene pores. Are those results consistent with the interpretation provided here? Related, could free energy barriers such as those extracted from those simulations be useful for informing the engineering models used in the present manuscript?

We thank the reviewers for their time and efforts to carefully review our manuscript and provide constructive feedback. We have revised the manuscript and have outlined our point-by-point response below. Reviewer comments appear in blue font. Changes made to the manuscript are highlighted in yellow.

We hope that the reviewers find the revised manuscript suitable for publication.

Reviewer #1 (Remarks to the Author):

Authors reported that water vapour transport is orders of magnitude higher (x80) than the liquid water transport. These observations for water vapour transport already made by few previous studies i.e. by Celebi et al (5×10^6 g m⁻² day⁻¹ for bilayer graphene with 200nm pores, Ref.39) and liquid water transport by osmotic pressure by Yang et al (5 g m⁻² day⁻¹ Pa⁻¹, ref31: and this value is in agreement with the value 0.6 g m⁻² day⁻¹ Pa⁻¹ reported in this draft by authors) through nanopores of monolayer graphene. From experimental point of view, report of water vapour transport through nanopores (<0.66nm) of single layer graphene is a new theme here. Then, authors invoked series resistance model to shed insights on the mechanisms of water and vapour transport through nanoscale confinement and they attributed to different transport regimes based on the higher flow rates of vapour than liquid water.

We appreciate the reviewer's positive comments on our study.

We agree that the report of water vapor transport through nanopores (<0.66 nm) of single layer graphene is a new theme in this manuscript but wish to clarify some additional points below:

- i. We note that Celebi et al. studied water vapor transport through 400 nm diameter pores in bilayer graphene (Ref. 40 in revised manuscript, DOI: 10.1126/science.124909), while we probe water vapor transport through sub-nanometer pores (<0.66 nm) in monolayer graphene. The difference in pore sizes is ~2-2.5 orders of magnitude and extremely important to note.
- ii. Yang et al. (in Ref. 31, DOI: 10.1126/science.aau532) reported liquid water transport through nanopores in monolayer graphene, but, they did not probe/report water vapor transport.

Hence, to the best of our knowledge, our study is one of the first to report on the difference in water and water vapor transport through nanopores (<0.66 nm) in monolayer graphene, which is highly relevant to many applications spanning several scientific disciplines.

Finally, we note the accuracy of our results/experiments/analysis is further validated by the broad agreement of our data on water transport with that obtained by Yang et al.'s data (DOI: 10.1126/science.aau532) but not by Surwade et al. (Ref. 30, DOI: 10.1038/nano.2015.37).

However, authors need to address the following comments to support their findings

The reviewer raises some interesting points and most of them pertain to the mechanisms of liquid and vapor permeation and transport modeling.

To fully address these concerns, we have performed molecular dynamics simulations of water vapor permeation through graphene nanopores of various sizes with carbon, hydrogen, and hydroxyl termination groups (See Fig. 3G, section S2, Table S2 and S3 in supporting information and Fig. S13-16).

We have further compiled a summary of some relevant, published molecular dynamics simulation data for liquid water permeation through graphene nanopores (see **Fig. 3G** and **Fig. S17**). Below, we copy the changes made to the main paper and supporting information to reflect the changes based on the molecular dynamics simulations and refer to it in the point-by-point response to the reviewer comments that follow (see below):

Revised text added to the manuscript:

Simulations have revealed a variety of nanoscale transport phenomena that occur during flow through graphene pores including velocity slip, viscosity changes under nanoconfinement, the formation of dense liquid layers along solid boundaries, adsorption of gas molecules on graphene, surface diffusion, and interactions between fluid molecules and terminal groups on the edges of the pore.^{15,33,60,65–70} The importance of such effects on transport rates was quantified by performing molecular dynamics simulations of water vapor permeation through various graphene nanopores (Figs. S13-S16, Tables S2 and S3) and by compiling published simulation data for liquid water transport through graphene nanopores (Fig. S17).^{15,33,60,71–74} In both cases, flow rate enhancements of up to a factor of ~ 3 compared to the simplified Knudsen effusion and Sampson flow models were calculated for some pores (Fig. 3G; details in Supporting Information Section S2). Although these nanoscale transport phenomena influence the precise flow rates through graphene nanopores,⁷⁵ the resulting factors of up to ~ 3 enhancements in both vapor and liquid flow rates are significantly smaller than the factor of ~ 80 difference measured between vapor and liquid transport rates. This order of magnitude difference is accounted for by the difference between the free molecular and near-continuum flow regimes in which vapor and liquid water molecules pass through the pores. This difference can be appreciated more simply by considering the analytical Knudsen effusion and Sampson flow models, although the molecular scale details would be necessary for more precise modeling.

Further sources of modeling error include uncertainty in the measured pore size distribution due to the limited sample size of graphene permeable pores that could be imaged and uncertainty in the precise nanopore size cut off at which POSS will plug the pore. This uncertainty may mask flow rate enhancements compared to our simple modeling if they occur. Due to this uncertainty, the relative difference in liquid and vapor transport rates, and the overall order of magnitude of permeance, is more meaningful than the precise values calculated in Fig. 3F. The small inherent permeance of the POSS has also been neglected. Nevertheless, these simple models quantitatively explain the large difference in permeance measured for water vapor and liquid water transport based on the measured pore size distribution. We note that while we have included the resistance of the support membrane (R_{supp}) in the modeling, omitting it in this case would only change flow rate predictions by $\sim 11\%$ for water vapor and $<0.1\%$ for liquid water. Further details on these transport models are provided in Supporting Information Section S1.

Fig. 3. Comparison of simulation models to measurements. A) Comparison of water vapor permeances between PCTE support and fabricated GMs under different mean relative humidity (RH) of 30% and 40%. B) Comparison of liquid water permeances between PCTE support and fabricated GMs under forward osmosis (FO) and reverse osmosis (RO). C) Permeance ratio of GM/PCTE for water vapor and liquid water. The GMs show very high water vapor permeance (~16.8%-17% of PCTE support) but very limited water permeance (~1.2% of PCTE support under FO and ~0.0086% of PCTE support under RO). All error bars indicate one standard deviation. D) Structure of a single support PCTE membrane pore with graphene suspended over it. E) Transport resistance model (P_H and P_L are the FO or RO pressures, or partial pressures, on either side of the membrane). F) Comparison of the transport model with the measured liquid

and vapor permeances. G) Comparison of liquid water^{15,33,60,71–74} and water vapor permeation coefficients from molecular dynamics simulations. Uncertainty bars for water vapor simulations show 95% confidence interval for a Poisson process. Legend format for liquid MD markers is “authors (year), pore terminal groups, water model [citation number]”. Legend format for vapor MD markers is “pore terminal groups, temperature”.

Revised text added to supporting information:

S2. Molecular dynamics simulations – transport modeling

In this paper, we have employed a simple analytical transport model to explain the significantly different transport rates measured in water vapor and liquid water permeation experiments. This modeling neglects many fascinating transport phenomena that have been uncovered in nanoflows. Here we more closely examine what impact such factors could have on the model predictions.

S2.1. Molecular dynamics simulations of water vapor transport across nanoporous graphene membranes: methodology

Molecular dynamics simulations were performed to estimate the discrepancy between the simple Knudsen effusion model employed and the actual water vapor flow rates through sub-nanometer pores in graphene. These simulations capture such effects as adsorption of vapor molecules on graphene, surface diffusion of vapor molecules along the membrane, and the atomic scale geometry of the pore and water molecules during crossings.

Molecular dynamics simulations were performed in LAMMPS (Large-scale Atomic/Molecular Massively Parallel Simulator)¹² and visualized in VMD (Visual Molecular Dynamics).¹³ A 44.2 Å × 59.5 Å graphene membrane spanned the width of the domain in two directions and was positioned in the center of the domain in the third direction (Fig. S13), the length of which was chosen to set the nominal pressure. Four identical, equally spaced holes were created in the graphene. The hole geometries simulated are shown in Fig. S14. Nine pores with carbon atoms on the pore rim were simulated (Fig. S14a-i, designated C-1 to C-9) matching those in Ref. ¹⁴, where similar permeance simulations were performed for other gas species. Four pores with hydrogen or hydroxyl groups on the pore rim were also simulated (Fig. S14j-m, designated F-1 to F-4) to increase the affinity of water vapor molecules to the pore rim and thereby capture the effects of high adsorption and test whether condensation could be induced within the pores. Periodic boundary conditions were imposed in all three directions. A second graphene sheet, this one without pores, was positioned parallel to the first at the end of the domain. This divides the domain into two equal volume reservoirs and prevents gas molecules from passing between the two except through the porous graphene membrane.

Fifty (50) water molecules were placed in the domain, half on one side of the porous graphene membrane and half on the other. A TIP4P water model¹⁵ was used with a long-range Particle-Particle Particle-Mesh (pppm) Coulombic solver (masses [g/mol]: 15.9994 (oxygen), 1.008 (hydrogen); charges [e]: -1.0484 (oxygen), 0.5242 (hydrogen); O-H bond length [Å]: 0.9572; H-O-H bond angle [degrees]: 104.52; distance from oxygen to massless charge [Å]: 0.1250; O-O Lennard Jones epsilon [kcal/mol]: 0.16275; O-O Lennard Jones sigma [Å]: 3.16435; Lennard Jones cutoff radius [Å]: 12; O-H and H-H Lennard Jones sigma and epsilon values were set to zero). The SHAKE algorithm was used to hold the water molecules rigid during the simulations. Membrane molecules were fixed in position for the duration of the simulation.

The Lennard-Jones potential parameters and charge used to model interactions between atoms are listed in Table S2. These parameters follow Ref. ¹⁶ based on Ref. ¹⁷ and ¹⁸. Interaction parameters between dissimilar

atoms were computed by Lorentz-Berthelot mixing rules. A cutoff radius of 12 Å was imposed on Lennard-Jones interactions.

Simulations were performed at 300 K, in the range measured in this study, and at 380 K, where higher vapor pressures can be simulated to increase the number of molecule crossings and reduce uncertainty in permeance. A time step of 1 fs was used. Initial water molecule velocities were drawn from a Maxwell-Boltzmann distribution at the prescribed temperature. Simulations were performed in the NVT ensemble using a Nosé-Hoover thermostat. Replicate simulations for the same pore size were initiated using different random initial velocities. Table S3 lists the number of replicate simulations and total simulation time for each pore and temperature.

The simulations were run at equilibrium, with both sides of the membrane at the same average pressure. Under these conditions, the water vapor is an ideal gas, so crossings in the forward and backward directions are approximately independent. The total number of crossings in both directions is recorded over time (e.g., Fig. S15). Molecule positions were recorded every 10,000 timesteps to calculate permeance. Crossings were counted each time a molecule passed from >10 Å away from the membrane on one side to >10 Å away from the membrane on the other side.

Permeation coefficient [molecules/s-Pa-pore] was calculated as,

$$\dot{N} = \frac{N}{2 \cdot 4 \cdot t \cdot P} \quad (\text{S22})$$

where N is the number of crossings counted in both directions, P [Pa] is the gas pressure, t [s] is the simulated time, the factor of 2 accounts for crossings in both directions, and the factor of 4 accounts for there being four identical pores in the membrane.

The gas pressure in Eq. S22 was calculated from the ideal gas law. Gas adsorption to the graphene membrane and graphene separator effectively reduce the bulk pressure in the gas for this small system. For each pore size, the pressure in Eq. S22 was corrected for this by calculating the average number of molecules in the volumes more than 10 Å away from the membrane and separator. Averaging was performed over all replicate simulations for a given pore size. This volume, and the average number of molecules in this volume, were used in the ideal gas law to calculate the bulk pressure. This correction reduced the pressure by ~1%. The average pressure over all replicate simulations for each pore type is reported in Table S3. In all simulations, the pressure corresponds to 55-56% relative humidity, based on saturation pressures of 3.6 kPa at 300 K and 129 kPa at 380 K.¹⁹

Flow rates are summarized in Fig. S16 and Table S3. Pore areas (A) were calculated using a hit-and-miss Monte Carlo method to find the area not within $D_m/\sqrt{2}$ of any membrane atom, where D_m is the Lennard-Jones diameter of a membrane atom. Effective pore diameter was calculated as the diameter of a circle with the same area, *i.e.*, $D = \sqrt{4A/\pi}$. This is the method for calculating an effective graphene pore diameter proposed by Sun et al.¹⁴ for gas transport, where the factor of $\sqrt{2}$ approximately accounts for gas molecule collisions with the pore rim that deflect the molecule into the pore.

S2.2. Molecular dynamics simulations of water vapor transport across nanoporous graphene membranes: results

The molecular dynamics simulation results in Fig. S16 provide a way to estimate the error introduced by using a simple Knudsen effusion model for water vapor transport through graphene nanopores. Fig. S16 presents pore permeation coefficient (\dot{N} [molecules/s-Pa-pore]), which is related to the mass flow rate (\dot{m}_{pore} [kg/s-pore]) by,

$$\dot{N} = \frac{\dot{m}_{pore}}{m \Delta P} \quad (S23)$$

In the case of Knudsen effusion, substituting Eq. S6 into S23 shows that,

$$\dot{N} = \frac{\pi}{4} D^2 \frac{1}{\sqrt{2\pi m k_B T}} \quad (S24)$$

This expression is plotted in Fig. S16 alongside the molecular dynamics results.

For the smallest carbon terminated pore simulated, steric hinderance reduces the permeance compared to the Knudsen effusion model by ~80%. For the other eight carbon terminated pores simulated, the Knudsen effusion model can under-predicted permeance by up to ~70%. However, whereas the Knudsen effusion model assumes ballistic crossings of gas molecules from one side of the pore to the other, without collision, up to ~50% of the crossings from the molecular dynamics simulations result from an adsorption pathway, in which adsorbed vapor molecules diffuse along the membrane to the pore. This is shown by the “direct permeance” markers in Fig. S16, which present the contribution of ballistic crossings to the total permeance. This was calculated by counting the number of crossings in which a gas molecule passed from >10 Å from the membrane on one side to >10 Å from the membrane on the other side within 20 ps. We note that this approach is expected to under-count direct crossings because some slower moving gas molecules will take more than 20 ps to travel the 20 Å. From the Maxwell-Boltzmann distribution, this fraction of molecules is estimated to be 21% at 300 K and 19% at 380 K. The uncertainty introduced by this approach is included in the error bars presented in Fig. S16 for direct permeance. For all but the smallest carbon terminated pore simulated, the direct permeance computed from the simulations is within 30% of the Knudsen effusion model.

The UV/ozone generated graphene pores in this study are expected to have functional groups passivating the pore rim. These groups could increase the affinity of water vapor molecules for the surface, increasing adsorptive transport, or potentially causing condensation of water vapor in the pore. To investigate this possibility, we performed simulations on four graphene pores with hydrogen or hydroxyl terminal groups (pores F-1 to F-4) at 2000 Pa nominal pressure. The resulting flow rates are summarized in Fig. S16. Whereas the carbon terminated pores had flow rates less than 1.7 times higher than predicted by the Knudsen effusion model, flow rates through functionalized pores were up to 2.5 times higher. A smaller fraction of the permeance of the functionalized pores results from direct crossings (Fig. S16), indicating that increased adsorption on the membranes is contributing to this higher permeance.

Although the permeation rate by the adsorption pathway is enhanced for the membranes with functionalized pores, the greater affinity for these membranes did not cause condensation of water vapor within the pores. The average number of water molecules within a cylinder with the diameter of the pore and extending 10 Å above and below the pore was less than 0.03 for all pores simulated. Animations of all simulations were reviewed and at no point did groups of water molecules collect around the pores.

S2.3. Summary of molecular dynamics liquid water permeation rates through graphene nanopores

To assess the error introduced by using a simple Sampson flow model for graphene nanopore liquid water permeance, we compiled the summary plot in Fig. S17 of various published molecular dynamics simulation results for liquid water permeation rates through graphene nanopores. This plot includes simulations of graphene pores with various terminal groups and using different water models, as identified in the caption. Although pore diameter definitions can vary, Fig. S17 presents the pore diameter stated in each source

paper to avoid introducing errors while reconstructing pore geometries. This may introduce some data scatter.

The permeation coefficient (\dot{N} [molecules/s-Pa-pore]) predicted by the Sampson model can be obtained by substituting \dot{m}_{pore} [kg/s-pore] from Eq. S17 into Eq. S23, resulting in,

$$\dot{N} = \frac{D^3 \rho}{24 \mu m} \quad (\text{S28})$$

This curve is plotted in Fig. S17 for comparison to the molecular dynamics data. In this calculation, values of $\rho = 1000 \text{ kg/m}^3$ and $\mu = 0.001 \text{ Pa}\cdot\text{s}$ were used.

To better align with molecular dynamics simulation data for liquid water permeance of graphene nanopores, Suk & Aluru¹¹ developed a fit to correct for factors such as slip, finite pore aspect ratio, and viscosity changes under nanoconfinement within the pore. In terms of permeation coefficient, their fit becomes,

$$\dot{N} = \frac{\pi \left[\left(\frac{D}{2} \right)^4 + 4\delta \left(\frac{D}{2} \right)^3 \right] \rho}{8 \mu L_h m} \quad (\text{S29})$$

where,

$$L_h = 0.27 \left(\frac{D}{2} \right) + 0.95 \times 10^{-9} \text{ m} \quad (\text{S30})$$

$$\delta = \frac{1.517 \times 10^{-19} \text{ m}^2}{D/2} + 0.205 \times 10^{-9} \text{ m} \quad (\text{S31})$$

$$\mu = \frac{8.47 \times 10^{-13} \text{ Pa s m}}{D/2} + 0.00085 \text{ Pa s} \quad (\text{S32})$$

The difference in predicted permeation rate between this model and the Sampson model for a 10 \AA pore diameter is 8%. The absolute error decreases for smaller pore diameters.

Factors such as slip and affinity for terminal groups on the pore rim can enhance the permeation rate compared to the Sampson model (Eq. S28). Fig. 3G shows up to ~ 3.2 times higher permeation rates compared to the Sampson model in some cases over the range $0 \text{ \AA} < D < 10 \text{ \AA}$, relevant here. Although, it should be noted that Prasad et al.²⁰ have also observed differences in liquid water permeation rates of nearly a factor of ~ 2 for the same pore due to different choices of water model used in the molecular dynamics simulation.

S2.4. Model deficiencies

The simple Knudsen effusion model for vapor transport and Sampson flow model for liquid lose accuracy for pores similar in size to fluid molecules. They fail to capture several important transport mechanisms.^{11,14,16,21-25} In liquid water, deviations from continuum transport occur at these pore sizes. Velocity slip, viscosity changes under nanoconfinement, and the formation of dense liquid layers near the membrane alter liquid flow rates. Similarly, the non-zero size of gas molecules and adsorption on the membrane alter vapor flow rates. The extent of these deviations are quantified by the molecular dynamics simulation results summarized in Figs. S15 and S16.

Fig. 3G provides a comparison of liquid water and water vapor flow rates through graphene nanopores from molecular dynamics simulations. Molecular dynamics simulations predict enhancements in both liquid water and water vapor flow rates in comparison to the Sampson and Knudsen effusion models by up to a

factor of ~ 3 . Although such differences are important in determining the precise permeance values, they do not account for the larger factor of ~ 80 difference between liquid and vapor transport rates. Fig. 3G illustrates that, for various graphene pore structures, water vapor permeance is orders of magnitude higher than liquid water permeance. This large difference results from the different flow regimes in which liquid water and water vapor transport occurs across the same graphene nanopore. Although the Knudsen effusion and Sampson flow models do not resolve the precise permeation rates, which are affected by the nanoscale details of the pore, they do capture this difference in flow regime. As such, they provide the correct order of magnitude of both liquid water and water vapor flow rates, and the approximate difference between the two.

In modeling the liquid water and water vapor permeation rate through the membranes in this study, the Knudsen effusion and Sampson flow models are employed. A difference in graphene permeance by a factor of ~ 3 resulting from the molecular scale details of the pore, would significantly change the predicted permeance. Most pores simulated show less deviation than this, but nevertheless, in modeling, this difference would be lost in the inaccuracy of estimating pore density. Of the 89 pores imaged by STEM in this study, only 16 were in the size range of that would be permeable to water and not plugged by POSS-polyamide. This limited sampling introduced uncertainty in both the pore size distribution and density. Enhancements in flow rates due to molecular scale details of the pore are of similar magnitude in liquid and vapor, so errors in pore density will affect the magnitude of liquid and vapor permeance without significantly changing the ratio between the two. In modeling, using a fitted pore density rather than the measured pore density could be an appropriate approach in light of this uncertainty. However, the measured pore density happened to provide quantitative agreement of the transport model to the measurements here (which provided a self-consistency check to the results and some validation for multi-experimental approach using the same membrane), so using a fitted pore density was not employed.

In addition to uncertainty in the pore size distribution and density, the exact pore structures, and their distribution, is unknown. This would be required for accurate molecular simulation of precise permeation rates. This creates challenges in more detailed modeling of the membranes measured in this study.

Despite these deficiencies, the Knudsen effusion and Sampson flow models serve the purpose of explaining the factor of ~ 80 difference between water vapor and liquid water flow rates as resulting primarily from the different flow regimes in which transport occurs. Although there are important aspects of transport that are uncertain, such as the terminal groups on the pore edge, and the precise interactions of the water molecules with the graphene, the modeling explains the measured trends quite well. While there are many fascinating transport phenomena that can occur through sub-nanometer pores in graphene that can affect transport rates, the magnitude of these effects is much smaller than the differences between liquid and vapor transport rates. Molecular dynamics simulations have found enhancements in flow rates due to nanoscale surface interactions by up to a factor of ~ 3 . However, these differences fall far short of the factor of ~ 80 difference between liquid and vapor transport rates measured. This difference is well explained by the different flow regimes in which liquid and vapor transport occurs through graphene nanopores, and is captured by simple Knudsen effusion and Sampson flow models.

Fig. S13. Water vapor permeance molecular dynamics simulation snapshot illustrating domain setup. Water molecules (red hydrogen atoms and blue oxygen atoms) on both sides of a graphene (grey carbon) membrane containing four identical pores. Another graphene sheet without pores is positioned on the left end of the domain to separate the upstream and downstream reservoirs at that periodic boundary.

Table S2. Potential parameters drawn from Ref. 16 based on Ref. 17 and 18. Lennard-Jones parameters shown are between atoms of the same type. Interaction parameters between atoms of different types are calculated by Lorentz-Berthelot mixing rules.

	C (sp2)	C _{COH}	H _{COH}	O _{COH}	C _{CH}	H _{CH}	O _{water}	H _{water}
ϵ (kcal/mol)	0.0859	0.0703	0	0.155	0.046	0.0301	0.16275	0
σ (Å)	3.3997	3.55	0	3.07	2.985	2.42	3.16435	0
Charge, q (e)	0	0.2	0.44	-0.64	-0.115	0.115	-1.0484	0.5242

Table S3. Summary of molecular dynamics simulation results for water vapor transport through graphene pores.

Pore designation	Effective pore diameter [Å]	Temperature [K]	Number of replicate simulations	Total simulated time [ns]	Total molecule crossings (sum in both directions)	Average pressure [kPa]	Permeation coefficient [molecule/Pa-s-pore]
C-1	2.06	380	5	250	32	70.4	227
C-2	3.04	380	5	250	543	70.5	3851
C-3	3.78	380	5	250	845	70.5	5990
C-4	4.39	380	5	250	1065	70.6	7544
C-5	4.93	380	5	250	1213	70.7	8576
C-6	5.49	380	5	250	1460	70.5	10347
C-7	6.00	380	5	250	1622	70.7	11475
C-8	6.90	380	5	238	1939	70.6	14435
C-9	7.59	380	4	200	1888	70.4	16709
C-9	7.59	300	39	585	213	1.98	22993
F-1	9.31	300	15	750	343	2.00	28637
F-2	7.67	300	15	750	319	1.99	26653
F-3	4.78	300	15	750	65	1.99	5430
F-4	3.45	300	15	750	98	1.99	8194

Fig. S14. Pore geometries used in molecular dynamics simulations. Designations: **a** C-1, **b** C-2, **c** C-3, **d** C-4, **e** C-5, **f** C-6, **g** C-7, **h** C-8, **i** C-9, **j** F-1, **k** F-2, **l** F-3, **m** F-4. Grey is carbon, red is hydrogen, blue is oxygen.

Fig. S15. Example time traces of total number of molecule crossings (sum of both directions). Each trace corresponds to a simulation for a different pore at 380 K, with the pore designation indicated on the right.

Fig. S16. Water vapor molecular permeation coefficients from molecular dynamics simulations. The total permeation coefficient (unfilled markers), which includes all molecule crossings, is plotted along with direct permeation coefficient (filled markers), which counts only those molecules that passed from $>10 \text{ \AA}$ away on one side of the membrane to $>10 \text{ \AA}$ away on the other side of the membrane in less than 20 ps.

Uncertainty bars for total permeance show the 95% confidence interval for a Poisson process. Uncertainty bars for direct permeance account for both the 95% confidence interval for a Poisson process and for an up to 21% (19%) under-estimation of the direct crossings at 300 K (380 K) due to the fraction of molecules moving slowly enough that 20 ps would not be sufficient time to directly cross a 20 Å gap. Simulations were performed at a relative humidity of 55 to 56%. Legend format for markers is “pore terminal groups, temperature.”

Fig. S17. Compilation of published molecular dynamics simulation predictions for water permeation through graphene nanopores.^{11,16,20,25–28} The pore diameter plotted is that stated in the corresponding paper. Legend format for markers is “authors (year), pore terminal groups, water model [citation number].” Also see Fig. 3G.

New references (16 in total) added to the manuscript:

Ref. 35 (ACS Appl. Nano Mater. 2020, 3, 11897, doi.org/10.1021/acsnm.0c02464) was cited when reporting water transport through nanopores in clay interlayers “For example, theoretical and experimental investigations of water transport through nanopores in material systems such as thin-film composite (TFC) membranes,^{5–8} carbon nanotubes (CNTs),^{19–21} zeolitic imidazolate frameworks (ZIFs),²² bio-mimetic channels,^{23–28} carbon nanomembranes (CNM),¹⁸ two-dimensional (2D) capillary devices,^{16,17,29} 2D membranes,^{30–34} clay interlayers,³⁵ among others”.

Ref. 41 (Annu. Rev. Chem. Biomol. Eng., 2016, 7, 533, doi.org/10.1146/annurev-chembioeng-080615-034455) was added to the manuscript “The introduction of angstrom-scale defects into the graphene lattice allows for the creation of nanopores in an atomically thin membrane,^{32,41,42} where the pore length ~ 0.34 nm represents the theoretical minimum material thickness and approaches the molecular length scale of water ~ 0.28 nm (van der Waals diameter)”.

Ref. 64 (J. Chem. Educ. 1969, 46, 358, doi.org/10.1021/ed046p358) was cited when adding Graham effusion “The support pore resistance is estimated from the equation for Knudsen diffusion⁶³ (originated from Graham effusion⁶⁴)”.

Ref. 65 (J. Phys. Chem. C 2019, 123, 21309, doi.org/10.1021/acs.jpcc.9b02178), Ref. 66 (Annu. Rev. Fluid Mech. 2021, 53, 377, doi.org/10.1146/annurev-fluid-071320-095958), Ref. 67 (Langmuir 2012, 28, 16671, doi.org/10.1021/la303468r), Ref. 68 (J. Heat Transfer 2021, 144, 022701, doi.org/10.1021/la303468r), Ref. 69 (Eur. J. Mech. - B/Fluids 2022, 94, 366, doi.org/10.1016/j.euromechflu.2022.03.012), and Ref. 70 (Langmuir 2014, 30, 675, doi.org/10.1021/la403969g) were added to the manuscript “Simulations have revealed a variety of nanoscale transport phenomena that occur during flow through graphene pores including velocity slip, viscosity changes under nanoconfinement, the formation of dense liquid layers along solid boundaries, adsorption of gas molecules on graphene, surface diffusion, and interactions between fluid molecules and terminal groups on the edges of the pore.^{15,33,60,65–70}”.

Ref. 71 (J. Memb. Sci. 2021, 630, 119331, doi.org/10.1016/j.memsci.2021.119331), Ref. 72 (Carbon 2017, 116, 120, doi.org/10.1016/j.carbon.2017.01.099), Ref. 73 (J. Mol. Liq. 2020, 301, 112478, doi.org/10.1016/j.molliq.2020.112478), and Ref. 74 (Phys. Chem. Chem. Phys. 2018, 20, 16005, doi.org/10.1039/C8CP00919H) were cited when compiling published simulation data for liquid water transport through graphene nanopores “The importance of such effects on transport rates was quantified by performing molecular dynamics simulations of water vapor permeation through various graphene nanopores (Fig. S13-S16, Table S2) and by compiling published simulation data for liquid water transport through graphene nanopores (Fig. S17).^{15,33,60,71–74}”.

Ref. 75 (Nature 2020, 588, 250–253, doi.org/10.1038/s41586-020-2978-1) was added to the manuscript “Although these nanoscale transport phenomena influence the precise flow rates through graphene nanopores,⁷⁵ the resulting factors of up to ~3 enhancements in both vapor and liquid flow rates are significantly smaller than the factor of ~80 difference measured between vapor and liquid transport rates”.

Ref. 87 (J. Comput. Phys. 1995, 117, 1, doi.org/10.1006/jcph.1995.1039) and Ref. 88 (J. Mol. Graph. 1996, 14, 33, doi.org/10.1016/0263-7855(96)00018-5) were cited when performing simulations “Molecular dynamics simulations were performed in LAMMPS⁸⁷ (<http://lammps.sandia.gov>) and visualized in VMD (Visual Molecular Dynamics)⁸⁸”.

1. Authors claim that pore sizes are <6.6 Angstroms. This could in principle mean there are 1 angstrom sized pores for instance, which is probably misleading. To be precise and clear, the authors could specify a range (max and min size of pores, probably 5 Å to 6.6 Å, as the polymerizations seems to close the pores bigger than 5 Å). This change has to be made in abstract and in the main text.

We thank the reviewer for this helpful suggestion. We have revised the abstract and main text accordingly with the range of pore size capable of water transport **~2.8-6.6 Å**.

2. Water condensation – in page 4, line150: The authors say “Here, we emphasize that the phase of water molecules transporting through the GMs is water vapour, because i) the dew point under the experimental conditions used (RH= 30% or 40%, 30 °C, and atmospheric pressure) is ~10-15 °C”. With such condition of high moisture content, there is also a very likely possibility of pore condensation. Especially as the pores are much smaller (which authors points out in the very next sentence), the condensation can happen at much

smaller humidity. Have the authors considered this possibility? How do they justify that there is no mixed conduction of water + vapour.

We thank the reviewer for this thoughtful comment, please allow us to clarify.

- i. The water vapor pressure in small-diameter capillaries is known to drop according to the Kelvin equation. For example, for a capillary with a 0.3 nm radius and a water contact angle of 80°, the condensation pressure is predicted to be ~55% of the saturation pressure. However, for sub-nanometer pores in a monolayer graphene, it remains unknown if the Kelvin equation can give a reasonable estimate.
- ii. Together with the question regarding the validity of all continuum descriptions in sub-nanometer confinement, the thinness of these pores (comparable to water molecular size) adds an additional level of uncertainty. Essentially, there is no capillary length (graphene thickness ~0.34 nm) but only entrance/exit (i.e., edges), and, in this scenario, it is difficult to define what would be a contact angle or even a droplet of water. Feng et al. studied the influence of graphene nanopore edges on the water evaporation via molecular dynamics (MD) simulations and demonstrated that water evaporation flux increases dramatically as the pore size decreases due to the reduced free energy barrier for evaporation at the edges. [Nanotechnology 2019, 30, 165401, DOI: 10.1088/1361-6528/aafcbd] If the low free energy barrier at the nanopore edges of monolayer graphene allows for fast water evaporation (from liquid to gas), then water vapor (from gas to gas) diffusion should be much easier.
- iii. There are no available and unambiguous methods to determine the state of water at this angstrom scale experimentally. But when running the water transport experiment, we have to pre-wet the graphene nanopores with ethanol followed by water. Otherwise, the water transport rate will be zero, consistent with observations of Celebi et al. (Ref. 40 in revised manuscript, DOI: 10.1126/science.124909) where they attributed this to capillarity for pore up to 2-2.5 orders of magnitude larger. These observations imply that the graphene membranes are hydrophobic and pore wetting is difficult to achieve even for the case of liquid phase water transport. We note that liquid water transport rates are much lower than water vapor transport rate. Hence, one would expect any condensation to yield measured transport rates closer to liquid water transport rate but this is not observed in our water vapor experiments. To the contrary we see much higher transport rates. We emphasize, the experimentally measured water and water vapor transport rates are distinctly different and this difference is reproducible after time intervals on the same membrane as well as across distinct membranes.
- iv. In the revised manuscript, we present molecular dynamics simulations for water vapor transport through graphene nanopores in the size range up to 10 Å (see Section 2 of supporting information). These include pores with carbon, hydrogen, and hydroxyl terminal groups at high vapor pressure to promote condensation. Simulation snapshots show no condensation and net permeation rates are within a factor of ~3 of the Knudsen effusion model.
- v. We note that our simplified modeling used the pore size distribution and density measured by STEM with no free parameters used to fit the experimental data. If condensation occurred in a significant fraction of the pores, we would expect the transport rates to be significantly altered. The fact that the vapor transport model agrees with the measurements is perhaps additional evidence that condensation is not occurring in a significant fraction of the pores.

Nonetheless, to fully address the reviewer's question, we not only performed simulations as detailed in Section 2 of the supporting information but also revised the manuscript as follows:

“Here, we **note** that the phase of water molecules transporting through the GMs is water vapor, because *i*) the dew point under the experimental conditions used ($\overline{RH} = 30\%$ or 40% , $30\text{ }^\circ\text{C}$, and atmospheric pressure) is $\sim 10\text{-}15\text{ }^\circ\text{C}$, *ii*) the pressure in the confined aperture is unlikely to reach the GPa range which is required to transform water vapor to liquid water at these conditions,⁹ *iii*) **prewetting was required to achieve liquid water permeation, thus indicating that the membrane pores are hydrophobic, and *iv*) our molecular dynamics simulations for water vapor transport through graphene nanopores (with carbon, hydrogen, and hydroxyl terminal groups) in the size range up to 10 \AA show no water condensation despite the simulated higher relative humidity of $\sim 55\%$ to promote condensation (see details in Supporting Information Section S2). Finally, a modest increase in WVTR observed with increasing $\overline{RH}\%$ for PCTE as well as GMs (Fig. 2B) indicates a transport mechanism inconsistent with water evaporation, since evaporative flux typically decreases with increasing RH (a trend that is opposite to the experimental observation).”**

Ref. 75 (Nature 2020, 588, 250–253, doi.org/10.1038/s41586-020-2978-1) was also added to the manuscript “**Although these nanoscale transport phenomena influence the precise flow rates through graphene nanopores,⁷⁵ the resulting factors of up to ~ 3 enhancements in both vapor and liquid flow rates are significantly smaller than the factor of ~ 80 difference measured between vapor and liquid transport rates”.**

3. For water vapour transport through nanopores of $<6.6\text{ \AA}$, water molecules will get absorbed inside the pore surface which will reduce the effective pore size by its dimension 3 \AA (roughly), therefore the effective pore diameter (free space) left for water vapour molecules (WVMs) will be $<3.3\text{ \AA}$ and within such a confinement regime it's reasonable to assume that WVMs will be bonded by hydrogen bonds and therefore molecules will not be free essentially, probably becomes bigger water droplet inside nanopore of such tight confinement regime with 30% - 40% RH. It has been observed/known that for graphene/graphene derivatives, such small pore membranes could trap water molecules for few days once it is exposed to water vapour. In such scenario, assumptions of free molecular transport probably might not a good starting point (using those corresponding equations).

We thank the reviewer for raising this hypothetical scenario, please allow us to elaborate.

Indeed, it has been reported that it could take up to several days for long graphite capillaries exposed to high RH to dry out the water completely. [Nature 2020, 588, 250, doi.org/10.1038/s41586-020-2978-1] However, here we are dealing with a distinctly different system/material geometry *i.e.* monolayer graphene with nanopores. As detailed in response above (point #2 to reviewer 1), we see clear differences in transport of water and water vapor through nanopores in graphene. Water vapor permeation is observed to be faster experimentally. In the hypothetical scenario of formation of liquid water in the pore we would expect slower transport rate. However, this is **not** observed in our experiment.

Nonetheless, to fully address the reviewer's question we provide the following response:

As the reviewer points out, it has been reported that it could take up to several days for long graphite capillaries exposed to high RH to dry out the water completely. [Nature 2020, 588, 250,

doi.org/10.1038/s41586-020-2978-1] However, when talking about monolayer graphene nanopores (rather than long graphite capillaries), it's uncertain what will happen between water molecules and monolayer graphene nanopore edges, since there is no capillary length. As articulated in the response to point 1 from reviewer #1, simulations [Nanotechnology 2019, 30, 165401, DOI: 10.1088/1361-6528/aafcbb] indicate water evaporation flux increases dramatically as the monolayer graphene nanopore size decreases due to the reduced free energy barrier for evaporation at the edges. If monolayer graphene nanopore edges enhance the water evaporation (from liquid to gas), then water vapor (from gas to gas) diffusion should be much easier.

To further explore this point, we performed molecular dynamics simulations of water vapor transport through graphene nanopores. The simulation details are provided in the revised **Supporting Information Section S2**. We simulated water vapor permeation through 13 graphene nanopores of less than 10 Å size with carbon, hydrogen, and hydroxyl terminal groups (see **Fig. 3G, Fig. S13-S17, Table S2 and S3**). Adsorption of water vapor molecules to the graphene was observed and quantified by calculating the pressure drop in the bulk gas due to increased gas molecule density near the surface. Permeance was separated into contributions from (1) bulk-to-bulk direct flux and (2) surface diffusion of adsorbed molecules. Adsorbed molecules provided a significant contribution to permeance, accounting for more than 50% of molecule crossings in the cases of hydrogen and hydroxyl terminated pores. However, simulation snapshots do not show condensation within the pores. The average number of molecules within a cylinder of the pore diameter and extending 10 Å above and below each pore was calculated. The maximum value over all pores was less than 0.03, indicating that there were not prolonged residence times of water molecules near the pore.

4. As a continuation of the above discussions, it might be useful to show $1/\sqrt{T}$ dependency to prove its free molecular transport of water vapour through nanopore (by comparing Eqn6 and Eqn 10 in the main draft), as only the proof (rough estimations) comes from those equations provided by authors.

We thank the reviewer for their suggestions. Please allow us to explain.

We agree with the reviewer and, conceptually, we could extract the dominant transport mode from the temperature (T) dependence of the transport rates. Unfortunately, the experimental measurements are not sensitive enough to resolve between different scaling laws. Specifically, the temperature range that we can probe is limited to 10-90 °C, which corresponds to a maximum change in transport rate of 12% based on the inverse square root T-dependence of free molecular transport. This is comparable to the experimental error in the measurements (~5-10% for repeated measurements at the same conditions). In addition to this limitation, the measured water vapor fluxes include the contribution of the boundary layer (BL) resistance at the two membrane surfaces, which is also temperature dependent (the bulk gas diffusivity scales with $T^{1.5}$, a stronger dependence than that of the effusion law). The BL resistance is non-negligible for high-flux membranes such as the GM membranes of this study. Quantification of (and correction for) this BL resistance is also subjected to some % error at each temperature condition which adds more uncertainty. Thus, the compounded uncertainties in the experimental values obtained with the MVTR method is too large to achieve an unambiguous determination of the temperature scaling for the measured water vapor fluxes.

5. A few factors that could enhance the water vapour transport: (a) Water evaporation on permeate side. It is possible that water evaporation flux happens from permeate side of the membrane for such tighter nanopores (in addition, the sweeping gas flows on the permeate side with lower RH) and thereby authors see much enhanced transport rate as compared to liquid water transport (evaporation driven enhanced flux). Authors need to address this point. A clear distinction is needed here to differentiate the water vapour

molecular transport or water evaporation occurring on the other side of the membrane as transport rate of water evaporation will also give orders of magnitude difference than the rate for liquid water transport.

As detailed above (response to points 2,3 for reviewer #1), the revised paper presents molecular dynamics simulations of water vapor transport through graphene (see supporting information section 2, Table 2,3 and S13-17, Fig. 3G). While these simulations show enhanced transport due to water vapor molecule adsorption on graphene, they do not show evidence of condensation within the graphene pores. “Finally, a modest increase in WVTR observed with increasing RH % for PCTE as well as GMs (Fig. 2B) indicates a transport mechanism inconsistent with water evaporation, since evaporative flux typically decreases with increasing RH (a trend that is opposite to the experimental observation).”

We appreciate reviewer #1’s interest in condensation as detailed in points #2, 3 and 5 but prefer to retain our current explanation of mechanisms in the manuscript that are effectively supported by our own data and modelling efforts and fully self-consistent.

(b) Enhanced interaction with functionalization groups present at the pore mouth (on the feeding site) due to UV/ozone etching. Pores created with etching method contains functional groups (FGs) attached to its pore edges which will modify the water vapour transport. The interactions between FGs with the water molecules would exert additional forces which will, in-turn, render the transport rate/mechanisms. These forces will be negligible when bulk liquid water comes in contact with those attached FGs, because those interactions for water vapour molecules will change their other degrees of freedom (rotational and translational motions, polarization). Therefore, direct comparisons between transport rates for water vapour molecules and liquid water without counting these effects into calculations might not shed more insights onto the transport mechanisms through such tighter confinements.

We thank the reviewer for this thoughtful comment and appreciate the opportunity to respond.

We agree that functional groups on the pore rim may increase the affinity of water molecules for the pore and influence transport rates. The molecular dynamics simulations presented in the revised paper include pores of different sizes with hydrogen and hydroxyl terminal groups to investigate these effects (Fig. S13-17, Table S2 and S3 and section 2 in supporting information). We further compiled a summary of literature data for liquid water transport through graphene nanopores with different terminal groups. The results are summarized in Fig. 3G.

Molecular dynamics simulation results show flow rate enhancements of up to a factor of ~3 for both liquid water and water vapor compared to the simple Knudsen effusion and Sampson flow models. Hydrogen and hydroxyl terminal groups lead to enhanced permeance through greater surface diffusion of adsorbed molecules. Similarly, the affinity of liquid molecules to molecules on the pore rim has been found to cause dense layers of water molecules to form near the graphene and within the pore, effectively altering the pore size, viscosity, and transport within the pore [Suk & Aluru, RSC Adv., 2013, doi.org/10.1039/C3RA40661J]. However, Fig. 3G illustrates that the combined effects of these nanoscale transport behaviors result in factors of up to ~3 enhancements in both water vapor and liquid water flow rates compared to the Knudsen effusion and Sampson flow models, respectively, which are significantly smaller than the factor of ~80 difference measured between vapor and liquid transport. Notably, this factor of ~80 difference between liquid water and water vapor permeation rates is also evident in the molecular dynamics simulation data in Fig. 3G and results from the difference in flow regime in which the two occur. Although the Knudsen effusion and Sampson flow models do not resolve the precise permeation rates, which are affected by the nanoscale details of the pore, they do capture this difference in flow regime. As

such, they provide the correct order of magnitude of both liquid water and water vapor flow rates, and the approximate difference between the two.

(c) Also, surface diffusion could enhance the water vapour transport rates. Therefore, counting all these above mentioned factors towards the enhancement of water vapour transport, the actual resistance could be similar in range for liquid water and water vapour transport.

We thank the reviewer for this detailed input and appreciate the opportunity to clarify.

Firstly, the measured transport rates of water and water vapor in our study are significantly different (factor of ~ 80) and our modelling effectively captures these aspects. Nonetheless, to fully address the reviewer's concern on mechanistic insights we performed additional molecular dynamic simulations and provide a response below:

We agree that the simple Knudsen effusion model does not capture surface diffusion of water vapor, which can contribute significantly to transport. To quantify this effect, we have separated the contributions of surface adsorption from direct, bulk-to-bulk transport in the water vapor permeance molecular dynamics simulations (Figs. S14 and S16) presented in the revised paper. For carbon terminated pores, surface diffusion accounted for $\sim 50\%$ of the total permeance, leading to a factor of up to ~ 1.7 enhancement in water vapor permeance. Although many of the permeating molecules do so by a different mechanism than the Knudsen effusion model is based on, it gives the correct order of magnitude of permeance. Surface diffusion accounts for an even greater fraction of the transport through hydrogen and hydroxyl terminated pores, raising the total permeance to up to ~ 2.5 times that predicted by the Knudsen effusion model.

Although surface diffusion and other nanoscale transport phenomena influence the precise flow rates through graphene nanopores, molecular dynamics simulations of various graphene nanopores indicate that their combined effects result in factors of up to ~ 3 enhancements in flow rates (Fig. 3G). This is significantly smaller than the factor of ~ 80 difference measured between water vapor and liquid water transport rates. Comparing molecular dynamics simulation results for vapor transport rates to liquid transport rates (Fig. 3G) shows that, even with the combined effects of the various nanoscale transport behaviors reported for graphene nanopores, the permeation rates of water vapor and liquid water still differ by a factor of ~ 80 . Furthermore, the order of magnitude of water vapor permeation is correctly predicted by the Knudsen effusion model and the order of magnitude of liquid water permeation is correctly predicted by the Sampson flow model. Although the molecular scale details would be necessary for more precise modeling, the Knudsen effusion and Sampson flow models capture the factor of ~ 80 difference between liquid water and water vapor flow rates resulting primarily from the difference in flow regime between the two.

6. The liquid bulk water (on feeding site) has high dielectric constant (KDC) ~ 80 . Also, we know that when water molecules pass through tighter confinements it has very low KDC (around ~ 2) as previously reported. In general, the faster transport of water molecules occur through nanochannel because of reduced KDC (associated with ordered/layered water molecules inside nanopore). Therefore, the observed less transport rate for liquid water could be due to the presence of other limiting steps.

We thank the reviewer for the comment.

It has not escaped our attention that the dielectric constant of water molecules decreases from ~ 80 (bulk) to ~ 2 when confining water between two atomically flat walls [Science 2018, 360, 1139, DOI: 10.1126/science.aat4191].

However, the material system we use is completely different *i.e.* nanopores in atomically thin graphene (no capillary length).

Further, the impact on liquid water transport rates due to changes in dielectric constant caused by water molecule reorientation under nanoconfinement should be captured by molecular dynamics simulations of liquid water permeation, at least to the accuracy with which the water model can capture these dynamics.

To estimate the importance of such effect, published molecular dynamics simulation data for liquid water transport through graphene nanopores were compiled and summarized in Fig. S17 and Fig. 3G. Note that these results account for not only molecule reorientation but also for factors such as velocity slip and viscosity changes under nanoconfinement. These data provide an estimate of the magnitude of deviation from the Sampson flow model possible in liquid water transport through graphene nanopores due to the molecular scale details of transport. Enhancements in liquid water transport rates compared to the Sampson flow model are within a factor of ~ 3 . This is far less than the factor of ~ 80 difference between measured liquid water and water vapor transport rates. While such effects alter the precise value of water permeance, molecular dynamics simulations indicate that they do not change the order of magnitude of liquid water transport rates through graphene nanopores.

7. Use of large molecule, glycerol ethoxylate as a draw solution:

(a) In a way, this large molecule is blocking the membrane pores from the permeate side, which can hinder the water from feed to pass freely to the other side. What is the rationale behind using such large molecule? Is this mainly the reason for lower liquid water permeation, as the flow is being hindered? The authors should do a control experiment where they use smaller molecule on the permeate side, and compare the water flow.

There appears to be a misunderstanding. Please allow us the opportunity to clarify.

In addition to forward osmosis (FO) we perform mechanical pressure driven reverse osmosis (RO, see Fig. 2G) and obtained similar liquid water permeance across our graphene membranes (Fig. 2H). RO is a clean, clear and unambiguous measurement and notably its results are in excellent agreement with FO. The RO measurement appears to have been inadvertently missed by the reviewer and we re-iterate and highlight it for the reviewer's benefit.

The rationale for using glycerol ethoxylate as the draw solution is its relatively large average molecular weight (~ 1000 , close to B12) and molecular diameter (~ 1.2 nm, close to B12), leading to negligible transport of the draw solute across the sub-nanometer scale pores in graphene membrane to the opposite side (feed side), to ensure accurate water transport measurement by maintaining accurate driving force (osmotic pressure). Using small molecules (like salts) as the draw solution, there is a chance that salts will diffuse from the permeate side into the feed side during forward osmosis causing changes to the driving force. To rule out the effect from glycerol ethoxylate, we also performed mechanical pressure driven reverse osmosis (RO, see Fig. 2G) and obtained similar liquid water permeance across our graphene membrane (Fig. 2H).

(b) It is nice that authors have compared the experimental values of water and vapour transport, with those estimated from classical Poiseuille equation and Knudsen. However, in the case of liquid water transport, the role of slip (and surface diffusion if any) is not considered, which is expected to affect the liquid water flow estimates (enhanced flow rate is expected due to slip).

We appreciate the reviewer's thoughtful comment.

Suk & Aluru [60] present a thorough molecular dynamics study of liquid water flow through graphene nanopores. They quantify the effects of slip and viscosity changes under nanoconfinement on transport rate. We have added discussion of their model accounting for slip and other effects in the revised paper. For a 10 Å diameter pore, their model predicts a deviation of only 8% from the Sampson flow model, with lower absolute difference at smaller diameters. Further enhancements, up to a factor of ~3 above the Sampson flow model, are possible in some functionalized pores as illustrated by the compilation of molecular dynamics simulation results in Fig. 3G, Fig. S17 and section 2 in supporting information of the revised paper. However, these differences remain far smaller than the factor of ~80 difference between measured liquid water and water vapor flow rates (as also detailed in earlier responses above).

(c) For liquid water measurements (FO), on permeate side there is already draw solution. Now if we don't put any draw solution, then in principle liquid water transporting via GMs will exist the pore via evaporation flux which could show similar rate as water vapour experiments show. In such scenario, it's not easy to distinguish transport mechanisms of water molecules through/inside the nanopores.

If we understood this comment correctly - there appears to be a hypothetical pervaporation (liquid on one side and another side exposed to air) experiment proposed by the reviewer and an explanation for the how such an experiment will not provide meaningful insight?

We agree that the proposed experiment is unhelpful, and this is exactly the reason we refrained from performing such experiments. Additionally, we foresee wetting issues in such experiments that could lead to erroneous results.

We note our experiments in this study are carefully defined and executed to provide meaningful insights. The FO, RO and vapor transport measurements are each carried out in different set-ups that are capable of measuring the intended physical parameters. Finally, we emphasize that despite these distinctly different set-ups used, we see remarkably consistent results for liquid water transport experiment via RO and FO which validates the accuracy and self-consistency of our study.

8. In Fig. 2, the data for FO and moisture transport for various GM samples is “remarkably” consistent as authors also pointed out. How is this feasible? The support membranes themselves might have some pore distribution range, so each GM sample may have slightly different number of effective pores. Additionally the method of pore creation, interfacial polymerization (IP) will lead to further distribution in the pore densities (and sizes) in each sample. This is also evident in Fig. 1, where the GM pore density is shown. How is it possible that four different samples show same flux, with such accuracy? Authors need to explain this, and give realistic distribution of effective pore density ranges per sample.

We thank the reviewer for noticing the remarkable consistency of our experiments. We are happy to elaborate.

We rigorously aim to minimize variations in experimental parameters during membrane fabrication and testing so we can successfully isolate and study scientific effects with minimal noise. For example, the PCTE supports used in this study are from the same batch/lot, which helps to avoid any batch-to-batch variation (~1-20%). We also use the same large piece 2 × 8 cm of CVD graphene grown on Cu foil throughout the study to minimize variations in graphene quality, intrinsic defects density, wrinkles etc. in the as-synthesized graphene.

Next, we use an isopropanol-assisted hot lamination method to transfer graphene to PCTE [Ref. 43, Nanoscale 2021, 13, 2825, doi.org/10.1039/D0NR07384A]. This method is extremely reliable, reproducible and allows for consistently high transfer yield ($\geq 95\%$ here). The UV/ozone treatment of all graphene membranes was done simultaneously in one batch using the exact same parameters. Finally, interfacial polymerization was also performed using the same batch of monomer solutions and chemicals (Ref. 32, Nano Lett. 2020, 20, 5951, doi.org/10.1021/acs.nanolett.0c01934).

After fabricating graphene membranes, we first performed the water vapor transport in Lawrence Livermore National Laboratory, and then mounted the same membranes into the customized diffusion cell system for liquid water transport measurements at Vanderbilt University using identical experimental conditions as detailed in the methods section. The graphene membranes showed consistent results for both water vapor and liquid water transport.

This careful and pain-staking attention to detail during experimentation is a particular strength of our study/approach and enables remarkably consistent results for a thorough study. Finally, the robust experimental techniques to minimize variability allows us to use representative membranes for nanopore imaging and extracting pore size distributions and is fully consistent with our own prior studies Ref. 32, Nano Lett. 2020, 20, 5951, doi.org/10.1021/acs.nanolett.0c01934.

9. Limit of detection: The measurement techniques for salt rejection and organic molecule rejection are based on conductivity and optical probe. However, the authors should mention the limit of detection/accuracy of the measurement techniques. If there is even a small amount of salt permeating through the pores, this might not be captured as it is diluting a larger volume of permeate solution. In the experimental section, the authors mentioned about normalization of concentration profiles of salts, however data should be presented for these control experiments. Also, the authors should present the limit of detection for their both conductivity and optical probe methods using impermeable samples (perhaps could be same graphene membranes without pores), and compare with their GM samples.

The main theme of the paper is on difference is water and vapor transport. Permeation of salts and small molecules through graphene is in the $\sim 0.5\text{-}3\%$ range compared to bare supports and is used to indicate sealing of larger pores in graphene membranes and is fully consistent with prior studies (Ref. 32, Nano Lett. 2020, 20, 5951, doi.org/10.1021/acs.nanolett.0c01934). We note that several groups including ours have used these experimental techniques extensively in the past and these are well-documented in the literature. (ACS Nano 2012, 6, 10130, doi.org/10.1021/nn303869m; Nano Lett. 2014, 14, 1234, doi.org/10.1021/nl404118f; Nano Lett. 2015, 15, 3254, doi.org/10.1021/acs.nanolett.5b00456; Nanoscale, 2017, 9, 8496, doi.org/10.1039/C7NR01921A; Adv. Mater. 2017, 29, 1700277, doi.org/10.1002/adma.201700277; Adv. Mater. 2017, 29, 1605896, doi.org/10.1002/adma.201605896; ACS Nano 2017, 11, 10042, doi.org/10.1021/acsnano.7b04299; ACS Appl. Mater. Interfaces 2018, 10, 10369, doi.org/10.1021/acsami.8b00846; Adv. Mater. 2018, 30, 1804977, doi.org/10.1002/adma.201804977; Nano Lett. 2020, 20, 5951, doi.org/10.1021/acs.nanolett.0c01934; Nanoscale, 2021, 13, 2825, doi.org/10.1039/D0NR07384A; Nat. Nanotechnol. 2021, 16, 989, doi.org/10.1038/s41565-021-00933-0; Sci. Adv. 2021, 7, eabg6263, DOI: 10.1126/sciadv.abg6263; Adv. Mater. 2022, 34, 2108940, doi.org/10.1002/adma.202108940)

Nonetheless to fully address the reviewer's concerns we provide the corresponding control experiments below.

The limits of detection/accuracy of the conductivity meter and UV-vis optical probe as specified by the manufacturer are:

Mettler Toledo SevenCompact S230 conductivity meter: resolution: 0.001 $\mu\text{S}/\text{cm}$, accuracy: $\pm 0.5\%$.

Agilent Cary 60 UV-Vis Spectrophotometer: resolution of wavelength: 1.5 nm, stray light: $<1\%$, photometric accuracy (Abs): $\pm 1\%$.

We calibrate the probes with salts (KCl and NaCl using conductivity meter) and small molecules (and L-Tr and B12 using UV-vis spectrometer) and design experiment to select appropriate ranges over which robust data can be acquired. Notably, we ensure measured values are within the detection limits of the probe (see Fig. R1-R2, Table R1 and R2 below). In other words, the lowest measured values are at least an order of magnitude higher than our lowest calibration value, thereby ensuring accurate experiments.

Next, we measured solute permeation through PCTE control and graphene membranes. The rate of permeation of each solute was computed via the slope of concentration change in the permeate side and the normalized flux was calculated by dividing the slope of the graphene membranes (GM) by that of the bare PCTE support (see Fig. R2). This is important conceptually since taking the slope allows us to use the rate of transport and avoid over reliance on a specific value of concentration, thereby minimizing errors.

Fig. R1. Calibration curves of A) KCl, B) NaCl, C) L-Tr and, D) B12 solutions. For C) and D), different UV-vis wavelengths were used for measuring the intensity differences of corresponding species: 710 nm for DI water (reference wavelength), 279 nm for L-Tr, and 360 nm for B12, respectively. Note a quadratic polynomial fit is used for UV-Vis data in C and D while a linear fit is used for conductivity data in A and B.

Table R1. Calibration data of KCl and NaCl. Notably, the calibration lower bounds are three orders of magnitude higher than the detection limit of conductivity meter to ensure accurate measurements.

Concentration of salt (mol/L)	Conductivity of KCl ($\mu\text{S}/\text{cm}$)	Conductivity of NaCl ($\mu\text{S}/\text{cm}$)
1×10^{-5}	3.940	4.332
2.5×10^{-5}	6.812	6.168
5×10^{-5}	10.930	9.710
1×10^{-4}	17.720	16.370
2.5×10^{-4}	41.100	35.980
5×10^{-4}	79.140	69.650
1×10^{-3}	153.1	132.7
2.5×10^{-3}	376.2	319.5
5×10^{-3}	740	641.8
1×10^{-2}	1445	1240
2.5×10^{-2}	3535	2994
5×10^{-2}	6840	5768
1×10^{-1}	13200	11220

Table R2. Calibration data of L-Tr and B12.

Concentration of solute (mol/L)	Intensity of L-Tr	Intensity of B12
1×10^{-7}	0.075	0.020
2.5×10^{-7}	0.076	0.022
5×10^{-7}	0.077	0.028
7.5×10^{-7}	0.080	0.035
1×10^{-6}	0.083	0.039
2.5×10^{-6}	0.084	0.075
5×10^{-6}	0.104	0.137
7.5×10^{-6}	0.117	0.191
1×10^{-5}	0.130	0.264
2.5×10^{-5}	0.212	0.651
5.0×10^{-5}	0.345	1.271
7.5×10^{-5}	0.478	1.797
1×10^{-4}	0.609	2.227

Fig. R2. Concentration changes of solutes (KCl, NaCl, L-Tr and B12) in the permeate side across PCTE control and graphene membrane (GM1). The slopes of KCl and NaCl were computed using data measured from 10 to 15 min, while the slopes of L-Tr and B12 were computed using data measured from 20 to 40 min.

Finally, we note that any high-quality monolayer graphene synthesized over centimeter scale areas (including single crystalline graphene - Adv. Mater. 2017, 29, 1605896, doi.org/10.1002/adma.201605896) is typically accompanied by intrinsic defects, wrinkles and transfer induced damages (Ref. 43, Nanoscale 2021, 13, 2825, doi.org/10.1039/D0NR07384A; Nano Lett. 2014, 14, 1234, doi.org/10.1021/nl404118f; Nano Lett. 2015, 15, 3254, doi.org/10.1021/acs.nanolett.5b00456; Nanoscale, 2017, 9, 8496, doi.org/10.1039/C7NR01921A; Adv. Mater. 2017, 29, 1700277, doi.org/10.1002/adma.201700277; Adv. Mater. 2017, 29, 1605896, doi.org/10.1002/adma.201605896) and will not form effective controls. However, given the lowest measured values are at least an order of magnitude higher than our lowest calibration value, we believe the data are adequate and reliable.

10. For the resistance model, the resistance for GMs is much larger than the PCTE support ($R_{NG} \gg R_{supp}$) for both the measurements, liquid water and water vapour transport measurements. The quantitative analysis could have been shown only by the resistance of GMs (R_{NG}) instead of invoking series-resistance model. Therefore, in order to justify the series resistance model (R_{supp} could have been removed) authors need to provide more information towards this.

We appreciate the reviewer's input here. We agree that R_{supp} could reasonably be neglected in the graphene membrane models here. For liquid water, the difference is less than 0.1%. However, because of the difference in flow rate scaling in the free molecular and continuum transport models, the support resistance accounts for 11% of the total flow rate resistance of the composite graphene membrane. This contribution will be much less than the factor of ~ 80 difference between liquid and vapor permeation rates and could reasonably be neglected to explain that difference. To clarify this point we have added the following statement in the revised paper:

“We note that while we have included the resistance of the support membrane (R_{supp}) in the modeling, omitting it in this case would only change flow rate predictions by $\sim 11\%$ for water vapor and $<0.1\%$ for liquid water.”

11. It is likely that vapour transport is higher than liquid, but whether it is as high as 80x, needs to be proved more cautiously. Especially, considering the measurement of liquid water in FO could be lower due to the large permeate molecule blocking the pores on the membrane (along with comments-5&7).

We thank the reviewer for cautiously acknowledging that the transport of water vapor might be higher than liquid water based on our results.

As already detailed our response to a similar question in point #7 by the reviewer, we re-iterate it again for the benefit of the reviewer:

In addition to forward osmosis (FO), we perform mechanical pressure driven reverse osmosis (RO, see Fig. 2G) and obtained similar liquid water permeance across our graphene membranes (Fig. 2H). RO is a clean, clear and unambiguous measurement and notably its results are in excellent agreement with FO. The RO measurement appears to have been inadvertently missed by the reviewer and we re-iterate and highlight it for the reviewer's benefit.

The rationale for using glycerol ethoxylate as the draw solution is its relatively large average molecular weight (~ 1000 , close to B12) and molecular diameter (~ 1.2 nm, close to B12), leading to negligible transport of the draw solute across the sub-nanometer scale pores in graphene membrane to the opposite side (feed side), to ensure accurate water transport measurement by maintaining accurate driving force (osmotic pressure). Using small molecules (like salts) as the draw solution, there is a chance that salts will diffuse from the permeate side into the feed side during forward osmosis causing changes to the driving force. To rule out the effect from glycerol ethoxylate blocking the pore etc. as suggested by the reviewer, we already performed mechanical pressure driven reverse osmosis (RO, see Fig. 2G) and obtained similar liquid water permeance across our graphene membrane (Fig. 2H).

Although, authors quantitatively showed the agreement between resistance model and their different transport regimes, the microscopic picture is still lacking here for water vapour transport through such tighter confinements. As mentioned in comment-1, such nanopores always trap water vapour molecules inside and stays longer once it is exposed to water vapour. Therefore the measurements after that (on exposed membrane) will not be free molecular diffusion transport. Also, there are other mechanisms (comment-3) which will complicate the water vapour transport.

We appreciate the reviewer's input.

To address concerns over the microscopic picture, we rely on molecular dynamics simulation results in the revised paper. We performed molecular dynamics simulations of water vapor transport through graphene

nanopores over the relevant size range to these experiments (see Fig. 3G, Fig. S13-S17, section 2 in supporting information and table S1 and S2). We compiled a summary of published molecular dynamics simulation results for liquid water permeation rates through graphene pores (Fig. S17 and Fig. 3G). These simulations capture effects such as velocity slip, viscosity changes under nanoconfinement, the formation of dense liquid layers along solid boundaries, adsorption or condensation of gas molecules on graphene or within pores, surface diffusion, and interactions between fluid molecules and terminal groups on the edges of the pore. Accounting for all of these effects, the molecular dynamics simulation results predict differences in liquid and vapor transport rates compared to the corresponding simple transport models by a factor of ~ 3 . Furthermore, condensation of water on the graphene or within the pore is not observed in the water vapor simulations. These factor of ~ 3 differences, while important in understanding precise transport rates, fall far below the factor of ~ 80 difference between liquid and vapor transport rates.

The molecular simulation results in Fig. 3G illustrate that, when accounting for the various nanoscale transport phenomena that influence liquid and vapor permeation through graphene nanopores, the order of magnitude of both water vapor and liquid water transport rates remain in line with the Knudsen effusion and Sampson flow models, respectively. We believe that Fig. 3G provides more convincing evidence that differences between the free molecular and near-continuum transport regimes for water vapor and liquid water flow are primarily responsible for the orders of magnitude difference in measured permeation rates. The molecular scale details of transport, while important for precise permeance calculations, are not the primary factor accounting for this difference.

Finally, we note that probing the molecular details of transport through graphene pores is a challenging experimental task. Phenomena predicted by simulations can be difficult to verify experimentally at this length scale, though this is an ongoing goal in graphene membrane research. One reason for this is experimental techniques to observe individual molecules transport through nanoscale pores in monolayer graphene and its interaction with the pore edge etc. are currently un-available to the scientific community and hence, well-beyond the scope of the current paper.

Reviewer #2 (Remarks to the Author):

Description of transport dynamics in nanofluidics requires two variables largely: length and lateral span (e.g., diameter) of a nanochannel. When both are quite larger than molecules passing the nanochannel, transport dynamics can be described by continuum theories. For nanochannels that puts confinement of one of the variables, transport is known to show abnormal diffusion (e.g., carbon nanotube of tiny diameters) or ultimate permeation (e.g., supernanometric graphene orifice). What has not yet been well explored is the case when both variables are limited to the size commensurable with transmitting molecules own sizes, a transport regime where experiments and molecular simulations are both rare. In this manuscript, Kidambi and colleagues are using state-of-the-art meticulous methods to obtain transport data in the unknown transport regime. Instead of varying the orifice size with producing obscure data, they took a simple but clear path of comparing transports of liquid and vapor for one (range of) pore size. In this way, the authors report that a subnanometric orifice of graphene allows transports of liquid water and water vapor (moisture) follows continuum theory and free molecular flow physics, respectively. They fabricated subnanometric graphene orifice samples according to their previous method, with confirming the orifice pore size around 0.66 nm via hindered diffusion of reference ions/salts. For liquid water, they employed forward osmosis and pressure-driven setups to confirm each other. For water vapor transmission rate, an elaborate experiment by use of dynamic moisture permeation cell. By comparing experimental data with basic transport theories, they add a data of liquid water transmission that agrees with a continuum hydrodynamic theory. The fact that vapor transmission follows effusion is not surprising, and yet how faster the vapor transmission is than the liquid water permeation for ca. 0.6-nm-wide graphene orifices might serve the research community as a useful data in the future.

After several corrections made to the manuscript, I recommend this manuscript for potential publication.

We thank the reviewer for extremely positive comments on our work and recommending it for publication in Nature Communications. We really appreciate the reviewer's framing and contextualization of our work – thank you!

Suggestions for minor revision:

MVTR (moisture vapor transport rate) should be changed to WVTR (water vapor transmission rate).

We thank the reviewer's helpful and important suggestion. We agree with the reviewer and changed all the MVTR (moisture vapor transport rate) to WVTR (water vapor transmission rate) in the manuscript and supplementary information.

“Knudsen effusion” should be replaced by “Graham effusion”.

We thank the reviewer's helpful suggestion. We agree with the reviewer and have changed “Knudsen effusion” to “**Knudsen effusion (originated from Graham effusion⁶⁴)**” in the revised manuscript. We also cited Ref. 64 (J. Chem. Educ. 1969, 46, 358, doi.org/10.1021/ed046p358), when citing Graham effusion.

Liquid water permeance data of Fig. 3F should be shown in log scale for clarity, perhaps in the supplementary information.

We thank the reviewer's helpful suggestion. We agree with the reviewer and re-plot Fig. 3F in logarithmic scale in the supplementary information Fig. S12. We also added a break in the Y-axis for Fig. 3F in the manuscript to show the data more clearly.

Fig. S12. Comparison of the transport model with the measured liquid and vapor permeances shown in Fig. 3F with a logarithmic Y-axis.

Fig. 3F. Comparison of the transport model with the measured liquid and vapor permeances.

Reference section needs polishing. Most of references from the Science magazine come with typos next to the journal name. Author names of ref. 64 contain typos.

We thank the reviewer for carefully reviewing all aspects and offering extremely constructive suggestions. We checked the references and revised it to ensure appropriate formatting.

Reviewer #3 (Remarks to the Author):

In this manuscript, the authors developed graphene membranes with pore size of less than 0.66 nm and transferred them to a polycarbonate support layer. The pore size larger than the specified dimension is repaired through size-selective interfacial polymerization. Transport of water in two states of vapor and liquid is measured through these pores. It has been shown that while liquid water transport through these pores is limited, the vapor flux is in the same order of other nanoporous materials. That is, these nanoporous membranes are selective in water vapor transport and exclude liquid water and larger molecules. These membranes could provide a new material paradigm for a wide range of applications including desalination and protective fabrics. I support publication of this work after addressing the following comments:

We thank the reviewer for the extremely positive comments on our work and recommending it for publication in Nature Communications.

1) The approximation of continuum fluid for 0.66 nm confined liquid does not seem correct. How do you define viscosity in these length scales? And what is the value? The authors need to conduct MD simulations for better understanding of liquid transport in these scales.

As suggested by the reviewer we performed molecular dynamic simulations.

Indeed, the continuum model for liquid water transport breaks down at this length scale. However, molecular dynamics simulation results indicate that the discrepancy between actual flow rates through graphene nanopores and the continuum model equation remain within a factor of ~ 3 (see Fig. 3G, Fig. S13-S17, supporting information section 2 and Table S1 and S2). Because many molecular dynamics studies have been published on liquid water flow through graphene nanopores, we rely on the literature for estimates of the discrepancy from continuum modeling for liquid water. In the revised paper, we provide a compilation of published molecular dynamics simulation results for liquid water transport through graphene nanopores in Fig. S17 (shown below). The precise difference will depend on the pore structure (geometry and functional groups on the edge of the pore), which is not known. Molecular dynamics simulations also show that a factor of ~ 2 difference in flow rate can be predicted for the same pore just by choosing a different water model in the simulation [73]. Nevertheless, these differences, while important for the precise transport rate, are far from the factor of ~ 80 difference between water vapor and liquid water transport rates measured in this study. The main source of this difference is the different transport regimes in which liquid and vapor molecules permeate the pore. This difference between near-continuum and free molecular flow rate scaling is emphasized in the revised paper. We believe that the added plot of liquid water molecular dynamics simulation results (Fig. S17) and associated discussion provide a more convincing justification for the simplified transport models used in the paper.

To quantify differences between the Knudsen effusion model and water vapor transport rates, we performed molecular dynamics simulations as suggested by the reviewer. This is presented in the revised supporting information section 2. The calculated permeation rates are summarized in Fig. S16 and compared to liquid permeation rates in Fig. 3G. Differences from Knudsen effusion prediction of up to a factor of ~ 3 are observed, similar to the enhancement predicted for liquid flow.

In the analytical transport models presented, the viscosity of bulk water (0.001 Pa-s) is used because that value gives agreement with molecular dynamics simulation data to within a factor of ~ 3 as seen in Fig. S17. However, the reviewer is correct that viscosity loses meaning at the nanoscale as there is not sufficient space for the number of collisions required to produce the diffusive transport of momentum quantified by

viscosity. Suk & Aluru [60] present a thorough molecular dynamics study of liquid water flow through graphene nanopores. They quantify the effects of slip and viscosity changes under nanoconfinement on transport rate. Their model accounting for slip and other effects is discussed in the revised paper. For a 10Å diameter pore, their model predicts a deviation of only 8% from the Sampson flow model, with lower absolute difference at smaller diameters.

Below, we provide the added discussion in the main paper and supporting information sections of the revised paper addressing these and other aspects of the transport mechanisms:

Revised text added to the manuscript:

Simulations have revealed a variety of nanoscale transport phenomena that occur during flow through graphene pores including velocity slip, viscosity changes under nanoconfinement, the formation of dense liquid layers along solid boundaries, adsorption of gas molecules on graphene, surface diffusion, and interactions between fluid molecules and terminal groups on the edges of the pore.^{15,33,60,65–70} The importance of such effects on transport rates was quantified by performing molecular dynamics simulations of water vapor permeation through various graphene nanopores (Figs. S13-S16, Tables S2 and S3) and by compiling published simulation data for liquid water transport through graphene nanopores (Fig. S17).^{15,33,60,71–74} In both cases, flow rate enhancements of up to a factor of ~3 compared to the simplified Knudsen effusion and Sampson flow models were calculated for some pores (Fig. 3G; details in Supporting Information Section S2). Although these nanoscale transport phenomena influence the precise flow rates through graphene nanopores,⁷⁵ the resulting factors of up to ~3 enhancements in both vapor and liquid flow rates are significantly smaller than the factor of ~80 difference measured between vapor and liquid transport rates. This order of magnitude difference is accounted for by the difference between the free molecular and near-continuum flow regimes in which vapor and liquid water molecules pass through the pores. This difference can be appreciated more simply by considering the analytical Knudsen effusion and Sampson flow models, although the molecular scale details would be necessary for more precise modeling.

Further sources of modeling error include uncertainty in the measured pore size distribution due to the limited sample size of graphene permeable pores that could be imaged and uncertainty in the precise nanopore size cut off at which POSS will plug the pore. This uncertainty may mask flow rate enhancements compared to our simple modeling if they occur. Due to this uncertainty, the relative difference in liquid and vapor transport rates, and the overall order of magnitude of permeance, is more meaningful than the precise values calculated in Fig. 3F. The small inherent permeance of the POSS has also been neglected. Nevertheless, these simple models quantitatively explain the large difference in permeance measured for water vapor and liquid water transport based on the measured pore size distribution. We note that while we have included the resistance of the support membrane (R_{supp}) in the modeling, omitting it in this case would only change flow rate predictions by ~11% for water vapor and <0.1% for liquid water. Further details on these transport models are provided in Supporting Information Section S1.

Fig. 3. Comparison of simulation models to measurements. A) Comparison of water vapor permeances between PCTE support and fabricated GMs under different mean relative humidity (RH) of 30% and 40%. B) Comparison of liquid water permeances between PCTE support and fabricated GMs under forward osmosis (FO) and reverse osmosis (RO). C) Permeance ratio of GM/PCTE for water vapor and liquid water. The GMs show very high water vapor permeance (~16.8%-17% of PCTE support) but very limited water permeance (~1.2% of PCTE support under FO and ~0.0086% of PCTE support under RO). All error bars indicate one standard deviation. D) Structure of a single support PCTE membrane pore with graphene suspended over it. E) Transport resistance model (P_H and P_L are the FO or RO pressures, or partial pressures, on either side of the membrane). F) Comparison of the transport model with the measured liquid

and vapor permeances. G) Comparison of liquid water^{15,33,60,71–74} and water vapor permeation coefficients from molecular dynamics simulations. Uncertainty bars for water vapor simulations show 95% confidence interval for a Poisson process. Legend format for liquid MD markers is “authors (year), pore terminal groups, water model [citation number]”. Legend format for vapor MD markers is “pore terminal groups, temperature”.

Revised text added to supporting information:

S2. Molecular dynamics simulations – transport modeling

In this paper, we have employed a simple analytical transport model to explain the significantly different transport rates measured in water vapor and liquid water permeation experiments. This modeling neglects many fascinating transport phenomena that have been uncovered in nanoflows. Here we more closely examine what impact such factors could have on the model predictions.

S2.1. Molecular dynamics simulations of water vapor transport across nanoporous graphene membranes: methodology

Molecular dynamics simulations were performed to estimate the discrepancy between the simple Knudsen effusion model employed and the actual water vapor flow rates through sub-nanometer pores in graphene. These simulations capture such effects as adsorption of vapor molecules on graphene, surface diffusion of vapor molecules along the membrane, and the atomic scale geometry of the pore and water molecules during crossings.

Molecular dynamics simulations were performed in LAMMPS (Large-scale Atomic/Molecular Massively Parallel Simulator)¹² and visualized in VMD (Visual Molecular Dynamics).¹³ A $44.2 \text{ \AA} \times 59.5 \text{ \AA}$ graphene membrane spanned the width of the domain in two directions and was positioned in the center of the domain in the third direction (Fig. S13), the length of which was chosen to set the nominal pressure. Four identical, equally spaced holes were created in the graphene. The hole geometries simulated are shown in Fig. S14. Nine pores with carbon atoms on the pore rim were simulated (Fig. S14a-i, designated C-1 to C-9) matching those in Ref. ¹⁴, where similar permeance simulations were performed for other gas species. Four pores with hydrogen or hydroxyl groups on the pore rim were also simulated (Fig. S14j-m, designated F-1 to F-4) to increase the affinity of water vapor molecules to the pore rim and thereby capture the effects of high adsorption and test whether condensation could be induced within the pores. Periodic boundary conditions were imposed in all three directions. A second graphene sheet, this one without pores, was positioned parallel to the first at the end of the domain. This divides the domain into two equal volume reservoirs and prevents gas molecules from passing between the two except through the porous graphene membrane.

Fifty (50) water molecules were placed in the domain, half on one side of the porous graphene membrane and half on the other. A TIP4P water model¹⁵ was used with a long-range Particle-Particle Particle-Mesh (pppm) Coulombic solver (masses [g/mol]: 15.9994 (oxygen), 1.008 (hydrogen); charges [e]: -1.0484 (oxygen), 0.5242 (hydrogen); O-H bond length [\AA]: 0.9572; H-O-H bond angle [degrees]: 104.52; distance from oxygen to massless charge [\AA]: 0.1250; O-O Lennard Jones epsilon [kcal/mol]: 0.16275; O-O Lennard Jones sigma [\AA]: 3.16435; Lennard Jones cutoff radius [\AA]: 12; O-H and H-H Lennard Jones sigma and epsilon values were set to zero). The SHAKE algorithm was used to hold the water molecules rigid during the simulations. Membrane molecules were fixed in position for the duration of the simulation.

The Lennard-Jones potential parameters and charge used to model interactions between atoms are listed in Table S2. These parameters follow Ref. ¹⁶ based on Ref. ¹⁷ and ¹⁸. Interaction parameters between dissimilar

atoms were computed by Lorentz-Berthelot mixing rules. A cutoff radius of 12 Å was imposed on Lennard-Jones interactions.

Simulations were performed at 300 K, in the range measured in this study, and at 380 K, where higher vapor pressures can be simulated to increase the number of molecule crossings and reduce uncertainty in permeance. A time step of 1 fs was used. Initial water molecule velocities were drawn from a Maxwell-Boltzmann distribution at the prescribed temperature. Simulations were performed in the NVT ensemble using a Nosé-Hoover thermostat. Replicate simulations for the same pore size were initiated using different random initial velocities. Table S3 lists the number of replicate simulations and total simulation time for each pore and temperature.

The simulations were run at equilibrium, with both sides of the membrane at the same average pressure. Under these conditions, the water vapor is an ideal gas, so crossings in the forward and backward directions are approximately independent. The total number of crossings in both directions is recorded over time (e.g., Fig. S15). Molecule positions were recorded every 10,000 timesteps to calculate permeance. Crossings were counted each time a molecule passed from >10 Å away from the membrane on one side to >10 Å away from the membrane on the other side.

Permeation coefficient [molecules/s-Pa-pore] was calculated as,

$$\dot{N} = \frac{N}{2 \cdot 4 \cdot t \cdot P} \quad (\text{S22})$$

where N is the number of crossings counted in both directions, P [Pa] is the gas pressure, t [s] is the simulated time, the factor of 2 accounts for crossings in both directions, and the factor of 4 accounts for there being four identical pores in the membrane.

The gas pressure in Eq. S22 was calculated from the ideal gas law. Gas adsorption to the graphene membrane and graphene separator effectively reduce the bulk pressure in the gas for this small system. For each pore size, the pressure in Eq. S22 was corrected for this by calculating the average number of molecules in the volumes more than 10 Å away from the membrane and separator. Averaging was performed over all replicate simulations for a given pore size. This volume, and the average number of molecules in this volume, were used in the ideal gas law to calculate the bulk pressure. This correction reduced the pressure by ~1%. The average pressure over all replicate simulations for each pore type is reported in Table S3. In all simulations, the pressure corresponds to 55-56% relative humidity, based on saturation pressures of 3.6 kPa at 300 K and 129 kPa at 380 K.¹⁹

Flow rates are summarized in Fig. S16 and Table S3. Pore areas (A) were calculated using a hit-and-miss Monte Carlo method to find the area not within $D_m/\sqrt{2}$ of any membrane atom, where D_m is the Lennard-Jones diameter of a membrane atom. Effective pore diameter was calculated as the diameter of a circle with the same area, *i.e.*, $D = \sqrt{4A/\pi}$. This is the method for calculating an effective graphene pore diameter proposed by Sun et al.¹⁴ for gas transport, where the factor of $\sqrt{2}$ approximately accounts for gas molecule collisions with the pore rim that deflect the molecule into the pore.

S2.2. Molecular dynamics simulations of water vapor transport across nanoporous graphene membranes: results

The molecular dynamics simulation results in Fig. S16 provide a way to estimate the error introduced by using a simple Knudsen effusion model for water vapor transport through graphene nanopores. Fig. S16 presents pore permeation coefficient (\dot{N} [molecules/s-Pa-pore]), which is related to the mass flow rate (\dot{m}_{pore} [kg/s-pore]) by,

$$\dot{N} = \frac{\dot{m}_{pore}}{m \Delta P} \quad (\text{S23})$$

In the case of Knudsen effusion, substituting Eq. S6 into S23 shows that,

$$\dot{N} = \frac{\pi}{4} D^2 \frac{1}{\sqrt{2\pi m k_B T}} \quad (\text{S24})$$

This expression is plotted in Fig. S16 alongside the molecular dynamics results.

For the smallest carbon terminated pore simulated, steric hinderance reduces the permeance compared to the Knudsen effusion model by ~80%. For the other eight carbon terminated pores simulated, the Knudsen effusion model can under-predicted permeance by up to ~70%. However, whereas the Knudsen effusion model assumes ballistic crossings of gas molecules from one side of the pore to the other, without collision, up to ~50% of the crossings from the molecular dynamics simulations result from an adsorption pathway, in which adsorbed vapor molecules diffuse along the membrane to the pore. This is shown by the “direct permeance” markers in Fig. S16, which present the contribution of ballistic crossings to the total permeance. This was calculated by counting the number of crossings in which a gas molecule passed from >10 Å from the membrane on one side to >10 Å from the membrane on the other side within 20 ps. We note that this approach is expected to under-count direct crossings because some slower moving gas molecules will take more than 20 ps to travel the 20 Å. From the Maxwell-Boltzmann distribution, this fraction of molecules is estimated to be 21% at 300 K and 19% at 380 K. The uncertainty introduced by this approach is included in the error bars presented in Fig. S16 for direct permeance. For all but the smallest carbon terminated pore simulated, the direct permeance computed from the simulations is within 30% of the Knudsen effusion model.

The UV/ozone generated graphene pores in this study are expected to have functional groups passivating the pore rim. These groups could increase the affinity of water vapor molecules for the surface, increasing adsorptive transport, or potentially causing condensation of water vapor in the pore. To investigate this possibility, we performed simulations on four graphene pores with hydrogen or hydroxyl terminal groups (pores F-1 to F-4) at 2000 Pa nominal pressure. The resulting flow rates are summarized in Fig. S16. Whereas the carbon terminated pores had flow rates less than 1.7 times higher than predicted by the Knudsen effusion model, flow rates through functionalized pores were up to 2.5 times higher. A smaller fraction of the permeance of the functionalized pores results from direct crossings (Fig. S16), indicating that increased adsorption on the membranes is contributing to this higher permeance.

Although the permeation rate by the adsorption pathway is enhanced for the membranes with functionalized pores, the greater affinity for these membranes did not cause condensation of water vapor within the pores. The average number of water molecules within a cylinder with the diameter of the pore and extending 10 Å above and below the pore was less than 0.03 for all pores simulated. Animations of all simulations were reviewed and at no point did groups of water molecules collect around the pores.

S2.3. Summary of molecular dynamics liquid water permeation rates through graphene nanopores

To assess the error introduced by using a simple Sampson flow model for graphene nanopore liquid water permeance, we compiled the summary plot in Fig. S17 of various published molecular dynamics simulation results for liquid water permeation rates through graphene nanopores. This plot includes simulations of graphene pores with various terminal groups and using different water models, as identified in the caption. Although pore diameter definitions can vary, Fig. S17 presents the pore diameter stated in each source

paper to avoid introducing errors while reconstructing pore geometries. This may introduce some data scatter.

The permeation coefficient (\dot{N} [molecules/s-Pa-pore]) predicted by the Sampson model can be obtained by substituting \dot{m}_{pore} [kg/s-pore] from Eq. S17 into Eq. S23, resulting in,

$$\dot{N} = \frac{D^3 \rho}{24 \mu m} \quad (\text{S28})$$

This curve is plotted in Fig. S17 for comparison to the molecular dynamics data. In this calculation, values of $\rho = 1000 \text{ kg/m}^3$ and $\mu = 0.001 \text{ Pa}\cdot\text{s}$ were used.

To better align with molecular dynamics simulation data for liquid water permeance of graphene nanopores, Suk & Aluru¹¹ developed a fit to correct for factors such as slip, finite pore aspect ratio, and viscosity changes under nanoconfinement within the pore. In terms of permeation coefficient, their fit becomes,

$$\dot{N} = \frac{\pi \left[\left(\frac{D}{2} \right)^4 + 4\delta \left(\frac{D}{2} \right)^3 \right] \rho}{8 \mu L_h m} \quad (\text{S29})$$

where,

$$L_h = 0.27 \left(\frac{D}{2} \right) + 0.95 \times 10^{-9} \text{ m} \quad (\text{S30})$$

$$\delta = \frac{1.517 \times 10^{-19} \text{ m}^2}{D/2} + 0.205 \times 10^{-9} \text{ m} \quad (\text{S31})$$

$$\mu = \frac{8.47 \times 10^{-13} \text{ Pa s m}}{D/2} + 0.00085 \text{ Pa s} \quad (\text{S32})$$

The difference in predicted permeation rate between this model and the Sampson model for a 10 \AA pore diameter is 8%. The absolute error decreases for smaller pore diameters.

Factors such as slip and affinity for terminal groups on the pore rim can enhance the permeation rate compared to the Sampson model (Eq. S28). Fig. 3G shows up to ~ 3.2 times higher permeation rates compared to the Sampson model in some cases over the range $0 \text{ \AA} < D < 10 \text{ \AA}$, relevant here. Although, it should be noted that Prasad et al.²⁰ have also observed differences in liquid water permeation rates of nearly a factor of ~ 2 for the same pore due to different choices of water model used in the molecular dynamics simulation.

S2.4. Model deficiencies

The simple Knudsen effusion model for vapor transport and Sampson flow model for liquid lose accuracy for pores similar in size to fluid molecules. They fail to capture several important transport mechanisms.^{11,14,16,21-25} In liquid water, deviations from continuum transport occur at these pore sizes. Velocity slip, viscosity changes under nanoconfinement, and the formation of dense liquid layers near the membrane alter liquid flow rates. Similarly, the non-zero size of gas molecules and adsorption on the membrane alter vapor flow rates. The extent of these deviations are quantified by the molecular dynamics simulation results summarized in Figs. S15 and S16.

Fig. 3G provides a comparison of liquid water and water vapor flow rates through graphene nanopores from molecular dynamics simulations. Molecular dynamics simulations predict enhancements in both liquid water and water vapor flow rates in comparison to the Sampson and Knudsen effusion models by up to a

factor of ~ 3 . Although such differences are important in determining the precise permeance values, they do not account for the larger factor of ~ 80 difference between liquid and vapor transport rates. Fig. 3G illustrates that, for various graphene pore structures, water vapor permeance is orders of magnitude higher than liquid water permeance. This large difference results from the different flow regimes in which liquid water and water vapor transport occurs across the same graphene nanopore. Although the Knudsen effusion and Sampson flow models do not resolve the precise permeation rates, which are affected by the nanoscale details of the pore, they do capture this difference in flow regime. As such, they provide the correct order of magnitude of both liquid water and water vapor flow rates, and the approximate difference between the two.

In modeling the liquid water and water vapor permeation rate through the membranes in this study, the Knudsen effusion and Sampson flow models are employed. A difference in graphene permeance by a factor of ~ 3 resulting from the molecular scale details of the pore, would significantly change the predicted permeance. Most pores simulated show less deviation than this, but nevertheless, in modeling, this difference would be lost in the inaccuracy of estimating pore density. Of the 89 pores imaged by STEM in this study, only 16 were in the size range of that would be permeable to water and not plugged by POSS-polyamide. This limited sampling introduced uncertainty in both the pore size distribution and density. Enhancements in flow rates due to molecular scale details of the pore are of similar magnitude in liquid and vapor, so errors in pore density will affect the magnitude of liquid and vapor permeance without significantly changing the ratio between the two. In modeling, using a fitted pore density rather than the measured pore density could be an appropriate approach in light of this uncertainty. However, the measured pore density happened to provide quantitative agreement of the transport model to the measurements here (which provided a self-consistency check to the results and some validation for multi-experimental approach using the same membrane), so using a fitted pore density was not employed.

In addition to uncertainty in the pore size distribution and density, the exact pore structures, and their distribution, is unknown. This would be required for accurate molecular simulation of precise permeation rates. This creates challenges in more detailed modeling of the membranes measured in this study.

Despite these deficiencies, the Knudsen effusion and Sampson flow models serve the purpose of explaining the factor of ~ 80 difference between water vapor and liquid water flow rates as resulting primarily from the different flow regimes in which transport occurs. Although there are important aspects of transport that are uncertain, such as the terminal groups on the pore edge, and the precise interactions of the water molecules with the graphene, the modeling explains the measured trends quite well. While there are many fascinating transport phenomena that can occur through subnanometer pores in graphene that can affect transport rates, the magnitude of these effects is much smaller than the differences between liquid and vapor transport rates. Molecular dynamics simulations have found enhancements in flow rates due to nanoscale surface interactions by up to a factor of ~ 3 . However, these differences fall far short of the factor of ~ 80 difference between liquid and vapor transport rates measured. This difference is well explained by the different flow regimes in which liquid and vapor transport occurs through graphene nanopores, and is captured by simple Knudsen effusion and Sampson flow models.

Fig. S13. Water vapor permeance molecular dynamics simulation snapshot illustrating domain setup. Water molecules (red hydrogen atoms and blue oxygen atoms) on both sides of a graphene (grey carbon) membrane containing four identical pores. Another graphene sheet without pores is positioned on the left end of the domain to separate the upstream and downstream reservoirs at that periodic boundary.

Table S2. Potential parameters drawn from Ref. 16 based on Ref. 17 and 18. Lennard-Jones parameters shown are between atoms of the same type. Interaction parameters between atoms of different types are calculated by Lorentz-Berthelot mixing rules.

	C (sp2)	C _{COH}	H _{COH}	O _{COH}	C _{CH}	H _{CH}	O _{water}	H _{water}
ϵ (kcal/mol)	0.0859	0.0703	0	0.155	0.046	0.0301	0.16275	0
σ (Å)	3.3997	3.55	0	3.07	2.985	2.42	3.16435	0
Charge, q (e)	0	0.2	0.44	-0.64	-0.115	0.115	-1.0484	0.5242

Table S3. Summary of molecular dynamics simulation results for water vapor transport through graphene pores.

Pore designation	Effective pore diameter [Å]	Temperature [K]	Number of replicate simulations	Total simulated time [ns]	Total molecule crossings (sum in both directions)	Average pressure [kPa]	Permeation coefficient [molecule/Pa-s-pore]
C-1	2.06	380	5	250	32	70.4	227
C-2	3.04	380	5	250	543	70.5	3851
C-3	3.78	380	5	250	845	70.5	5990
C-4	4.39	380	5	250	1065	70.6	7544
C-5	4.93	380	5	250	1213	70.7	8576
C-6	5.49	380	5	250	1460	70.5	10347
C-7	6.00	380	5	250	1622	70.7	11475
C-8	6.90	380	5	238	1939	70.6	14435
C-9	7.59	380	4	200	1888	70.4	16709
C-9	7.59	300	39	585	213	1.98	22993
F-1	9.31	300	15	750	343	2.00	28637
F-2	7.67	300	15	750	319	1.99	26653
F-3	4.78	300	15	750	65	1.99	5430
F-4	3.45	300	15	750	98	1.99	8194

Fig. S14. Pore geometries used in molecular dynamics simulations. Designations: **a** C-1, **b** C-2, **c** C-3, **d** C-4, **e** C-5, **f** C-6, **g** C-7, **h** C-8, **i** C-9, **j** F-1, **k** F-2, **l** F-3, **m** F-4. Grey is carbon, red is hydrogen, blue is oxygen.

Fig. S15. Example time traces of total number of molecule crossings (sum of both directions). Each trace corresponds to a simulation for a different pore at 380 K, with the pore designation indicated on the right.

Fig. S16. Water vapor molecular permeation coefficients from molecular dynamics simulations. The total permeation coefficient (unfilled markers), which includes all molecule crossings, is plotted along with direct permeation coefficient (filled markers), which counts only those molecules that passed from $>10 \text{ \AA}$ away on one side of the membrane to $>10 \text{ \AA}$ away on the other side of the membrane in less than 20 ps.

Uncertainty bars for total permeance show the 95% confidence interval for a Poisson process. Uncertainty bars for direct permeance account for both the 95% confidence interval for a Poisson process and for an up to 21% (19%) under-estimation of the direct crossings at 300 K (380 K) due to the fraction of molecules moving slowly enough that 20 ps would not be sufficient time to directly cross a 20 Å gap. Simulations were performed at a relative humidity of 55 to 56%. Legend format for markers is “pore terminal groups, temperature.”

Fig. S17. Compilation of published molecular dynamics simulation predictions for water permeation through graphene nanopores.^{11,16,20,25–28} The pore diameter plotted is that stated in the corresponding paper. Legend format for markers is “authors (year), pore terminal groups, water model [citation number].” Also see Fig. 3G.

2) Is it only the pore dimension that matters here? Or the interaction of water molecules with walls plays a role? For example, could one achieve the same performance through Zeolite structures?

We thank the reviewer raising this point.

Permeation rate is primarily dependent on pore diameter, as seen by the increasing trend in permeance with pore diameter seen in the molecular dynamics summaries in Fig. S16-S17. However, you are correct that interaction of the water molecules with the walls do play a role. In Fig. S16 of the revised paper, we separate the contributions of surface diffusion by adsorbed gas molecules from the total permeance. Adsorption accounts for ~50% of water vapor transport through carbon terminated pores and a greater fraction for hydrogen and hydroxyl terminated pores. We further summarize molecular dynamics simulation data for liquid water transport through graphene nanopores with different terminal groups on the pore edge in Fig. S17. The affinity of liquid molecules to molecules on the pore rim has been found to cause dense layers of water molecules to form near the graphene and within the pore, effectively altering the pore size, viscosity, and transport within the pore [Suk & Aluru, RSC Adv., 2013, doi.org/10.1039/C3RA40661J]. Fig. S17 indicates that these can lead to flow rate enhancements by a factor of up to ~3 compared to the analytical Sampson flow model.

For the question regarding “could one achieve the same performance through Zeolite structures?”, we acknowledge our limited expertise with zeolite structures and refrain from speculating on it since it is well beyond the scope of the current paper.

3) How does temperature affect the vapor transport through these pores? Does interaction with the walls affect the mass flux. In the free molecular transport model, the wall interactions are not taken into account.

We agree with the reviewer that temperature could affect the water vapor transport through these nanopores. Conceptually, we could extract the dominant transport mode from the temperature (T) dependence of the transport rates. Unfortunately, the experimental measurements are not sensitive enough to resolve between different scaling laws. Specifically, the temperature range that we can probe is limited to 10-90 °C, which corresponds to a maximum change in transport rate of 12% based on the inverse square root T-dependence of free molecular transport. This is comparable to the experimental error in the measurements (~5-10% for repeated measurements at the same conditions). In addition to this limitation, the measured water vapor fluxes include the contribution of the boundary layer (BL) resistance at the two membrane surfaces, which is also temperature dependent (the bulk gas diffusivity scales with $T^{1.5}$, a stronger dependence than that of the effusion law). The BL resistance is non-negligible for high-flux membranes such as the GM membranes of this study. Quantification of (and correction for) this BL resistance is also subjected to several % error at each temperature condition. Thus, the compounded uncertainties in the experimental values obtained with the MVTR method is too large to achieve an unambiguous determination of the temperature scaling for the measured water vapor fluxes.

We agree that interaction of water vapor molecules with graphene can affect transport rates. In particular, molecular dynamics simulations have shown that gas molecule adsorption on graphene and surface diffusion toward the pore contribute significantly to the overall gas permeance of graphene nanopores [69].

To quantify this effect for water vapor, we present molecular dynamics simulations of water vapor transport through 13 graphene nanopores in the revised paper. The results are summarized in Fig. S16 and S17. Surface diffusion of adsorbed water vapor molecules accounts for more than 50% of permeance for some pores, as noted in the response to comment 2. Fig. S16 of the revised paper plots the predicted Knudsen effusion rate at 300 K and 380 K along with molecular simulation results for water vapor permeation at both temperatures. The results are consistent with the temperature scaling of this model, to within the uncertainty in the calculated flow rates. However, at low temperatures, it is possible that stronger water molecule adsorption to the graphene may alter this temperature scaling.

The simulation results in Fig. S16 and S17 show that nanoscale transport phenomena including surface diffusion on the graphene can lead to enhancements in water vapor flow rates by up to a factor of ~3 compared to the Knudsen effusion model. Similar enhancements in liquid water flow by up to a factor of ~3 compared to the Sampson flow model are observed in molecular dynamics simulation data in Fig. S17 resulting from the nanoscale details of transport.

The point we want to highlight is that these factor of ~3 differences are significantly smaller than the factor of ~80 difference measured between vapor and liquid transport rates. This large difference between liquid and vapor permeation rates primarily results from difference in the near-continuum and free-molecular transport regimes in which the two occur. This difference is captured by the simple analytical transport models. Although the molecular scale details would be necessary for more precise modeling, the Knudsen effusion and Sampson flow model capture the factor of ~80 difference between liquid water and water vapor flow rates resulting primarily from the difference in flow regime between the two.

4) How do you ensure perfect attachment of the Graphene membrane to the polycarbonate and no leaking through the imperfection between these two?

The reviewer raises an interesting question. Please allow us to clarify.

Graphene was first transferred to polycarbonate track etched (PCTE) support via isopropanol-assisted hot lamination method, which allows for very high transfer yield [Ref. 43, *Nanoscale* 2021, 13, 2825, doi.org/10.1039/D0NR07384A] (more than 95% here). Next, the PCTE/graphene membrane was subjected to UV/ozone treatment and was baked at 105 °C for 12 hours to remove water molecules trapped at the PCTE/graphene interface to ensure the membrane is completely dry before performing interfacial polymerization (IP). The removal of water molecules and adhesion of graphene to the PCTE support is necessary to limit and confine the IP to form within the PCTE support pores (*Nano Lett.* 2015, 15, 3254, doi.org/10.1021/acs.nanolett.5b00456; *Nanoscale*, 2017, 9, 8496, doi.org/10.1039/C7NR01921A; *Adv. Mater.* 2017, 29, 1700277, doi.org/10.1002/adma.201700277; *ACS Nano* 2017, 11, 10042, doi.org/10.1021/acsnano.7b04299; *Nano Lett.* 2020, 20, 5951, doi.org/10.1021/acs.nanolett.0c01934; *Nanoscale*, 2021, 13, 2825, doi.org/10.1039/D0NR07384A). As articulated in our own prior work (*Nanoscale*, 2017, 9, 8496, doi.org/10.1039/C7NR01921A; *Adv. Mater.* 2017, 29, 1700277, doi.org/10.1002/adma.201700277; *Nano Lett.* 2020, 20, 5951, doi.org/10.1021/acs.nanolett.0c01934; *Nanoscale*, 2021, 13, 2825, doi.org/10.1039/D0NR07384A). If we don't remove those water molecules, the solutions for IP process can wick into the interface between graphene and PCTE and form a continuous dense layer of IP with permeance properties similar to PCTE+IP membranes.

Reviewer #4 (Remarks to the Author):

Differences in water and vapor transport through Angstrom-scale pores in atomically thin membranes

P. Cheng et al.

Summary

The Authors report an arsenal of experimental results conducted for water transport through single layer graphene membranes, with controlled defects. The defects lead to rather small pores, which are small enough for preventing, e.g., salt ions transport. Significant characterisation is conducted for these materials, in order to interpret experimental data for transport. The results show significant differences between liquid vs gaseous transport. To enhance the interpretation of the experiments, macroscopic models are used, in which the pore sizes and membrane thickness are used as parameters. The models provide data in good agreement with the experiments, suggesting that the interpretation of the experimental data is reliable.

Recommendation

This is a very well written paper, with many relevant references. The Authors place the work in the context of recent contributions, and clearly identify some discrepancies in the literature. The results are frequently interpreted referencing to simulation results, also from the literature, and the presentation seems to be consistent. The subject matter is of high practical importance, and the experiments seem to be conducted convincingly.

Thus, I recommend publication of this piece of work once the Authors have addressed my comments below.

We thank the reviewer for the extremely positive comments on our work and recommending it for publication in Nature Communications. In particular, we appreciate the positive comments on the number of experimental techniques, characterization and interpretation of transport data.

Details

The Authors seem to have made significant advancements along the lines of producing graphene membranes of macroscopic size. One 2016 Annual Review of Chemical and Biomolecular Engineering (Joly et al.) on the carbon-water interface suggested that this was a practical hurdle that needs to be overcome. Could the Authors comment on how large membranes they would be able to build?

We thank the reviewer for the positive comment and for highlighting the challenges articulated in Annu. Rev. Chem. Biomol. Eng., 2016, 7, 533, doi.org/10.1146/annurev-chembioeng-080615-034455.

In the current paper we demonstrate centimeter-scale graphene membranes but in earlier work we have shown roll-to-roll processes for synthesizing graphene membranes (Appl. Mater. Interfaces 2018, 10, 10369, doi.org/10.1021/acsami.8b00846)

We further note the fabrication methods we used in the paper are scalable. For example, 1) the CVD process used to synthesize graphene is in principle only limited by the size of the Cu catalyst substrate and reactor designs have allowed for roll-to-roll synthesis (Appl. Mater. Interfaces 2018, 10, 10369, doi.org/10.1021/acsami.8b00846). 2) The process to transfer graphene to PCTE support via isopropanol-assisted hot lamination is compatible with roll-to-roll manufacturing (Nanoscale 2021, 13, 2825,

doi.org/10.1039/D0NR07384A) and can be scaled to meet membrane sizes. 3) UV/ozone treatment is compatible with roll-to-roll fabrication and finally, 4) the interfacial polymerization (IP) used is also very scalable, and is typically used to fabricate large area membranes.

To fully address the reviewer's comment we have cited the mentioned article [Annu. Rev. Chem. Biomol. Eng., 2016, 7, 533, doi.org/10.1146/annurev-chembioeng-080615-034455] as Ref. 41. We note that although we demonstrate centimeter scale membranes, the approaches and fabrication techniques used can in principle be scaled to meter-scale areas relevant for practical applications.

The last sentence of the abstract, in my opinion, needs to specify what molecules larger than water have been blocked by the membranes being used here.

We thank the reviewer for the suggestion. We agree with the reviewer and have revised the last sentence of the abstract by specifying the salts and small organic molecules tested: KCl, NaCl, L-tryptophan and vitamin B12.

“We demonstrate centimeter-scale atomically thin graphene membranes with up to an order of magnitude higher water vapor transport rate ($\sim 5.4\text{-}6.1 \times 10^4 \text{ g m}^{-2} \text{ day}^{-1}$) than most commercially available ultra-breathable protective materials while effectively blocking hydrated salt ions ($\text{K}^+ \sim 0.662 \text{ nm}$, $\text{Cl}^- \sim 0.664 \text{ nm}$, and $\text{Na}^+ \sim 0.716 \text{ nm}$) and small organic molecules (L-tryptophan $\sim 0.7\text{-}0.9 \text{ nm}$ and Vitamin B12 $\sim 1\text{-}1.5 \text{ nm}$).”

In the introduction, the Authors refer to relevant literature for liquid vs. vapour water transport through membranes. There is a recent work by Tuan Ho and colleagues at Sandia who investigated similar differences, although through a clay membrane (if memory serves me correctly). They also report large differences in transport rates depending on the experimental conditions. I think that is a good reference to contrast different transport mechanisms.

We thank the reviewer for the constructive input and have cited the work in the introduction as Ref. 35. [ACS Appl. Nano Mater. 2020, 3, 11897, doi.org/10.1021/acsnm.0c02464]

The graphene group in Manchester produces graphene membranes for a variety of applications. Am I correct to interpret that the main difference is that the membrane used here is a single layer graphene, while those in Manchester tend to be multiple graphene layers deposited on each other? Could the Authors comment on how the results presented would change, in their opinion, if the ‘stack of graphene layers’ was used instead of the single graphene membrane?

We report on monolayer/single layer graphene membranes that are distinctly different from multi-layer lamellar membranes (see Figure below, Ref. 39, Adv. Mater. 2018, 1801179, doi.org/10.1002/adma.201801179).

Transport in multi-layer lamellar membranes occurs via gaps between the layers as well as defects in each of the layers. Transport in monolayer/single layer membranes occurs via nanopores in the single layers.

In our opinion the results would be very different between monolayer and multi-layer membranes since the transport pathways are very different between the two material systems. We refer the reviewer to several excellent studies on multi-layer membranes (Nature 2018, 559, 236, doi.org/10.1038/s41586-018-0292-y; Nat. Nanotechnol. 2017, 12, 546, doi.org/10.1038/nnano.2017.21; Nat. Mater. 2017, 16, 1198, doi.org/10.1038/nmat5025)

The Authors compare the high flow rate in their membrane to values obtained for commercial MVTR samples, which have larger pores. I think that for this comparison to be more convincing, the Authors should also report the surface density of the pores in the two substrates.

We thank the reviewer for this suggestion. Unfortunately, the commercial membranes do not provide surface density of pores.

In our study, we present the water vapor and liquid water transport results with and without accounting for the porosity of PCrTE (~9.4%) supports shown in the main manuscript (Fig. 2) and supporting information (Fig. S2 and S5). Specifically, we also compared our graphene membrane with commercial breathable materials in Fig. S2B without accounting for PCrTE support porosity *i.e.* as measured value of a membrane that is supported on a ~9.4% porosity substrate.

Fig. S2. (A) Measured WVTRs through PCTE and the fabricated GMs under different mean relative humidity (30% and 40%). The measured WVTR is calculated without accounting for ~9.4% porosity of PCTE supports. (B) WVTRs of the fabricated GM without accounting for ~9.4% porosity of PCTE supports and some commercial breathable materials¹ (ePTFE, eVent, Neoshell, Gore-Tex, and Sympatex) measured under 30% RH with a constant RH gradient of 50% across the membrane at 30 °C.

Fig. S5. A) Measured water flux across PCTE and the fabricated GMs under forward osmosis. B) Measured water flux across PCTE and the fabricated GM under reverse osmosis. Note water fluxes are calculated without accounting for ~9.4% porosity of PCTE supports.

To study vapor transport, the Authors chose conditions of 30-40% relative humidity. There are a number of simulation studies conducted using the grand canonical Monte Carlo formalism (see, e.g., the group of Gubbins) that investigated water sorption in carbon pores of various geometries. Could the Authors comment on whether at the conditions chosen here water confined in the graphene pores is expected to be high- or low- density, thus reminding of liquid vs vapor water?

Gubbins reported a series of water adsorption in carbon pores with various geometries via the grand canonical Monte Carlo simulations. [J. Phys. Chem. 1996, 100, 1189, doi.org/10.1021/jp952233w], [Langmuir 1999, 15, 2, 533, doi.org/10.1021/la9805950], [Langmuir 2002, 18, 5438, doi.org/10.1021/la0118560], [Langmuir 2003, 19, 8583, doi.org/10.1021/la0347354]. Those work are of very high importance to this field and throw new light on the adsorption mechanism for water. The main difference in this paper is we used monolayer graphene nanopores rather than long graphite capillaries, carbon slits or various activated carbon geometries.

To further explore this point, we performed molecular dynamics simulations of water vapor transport through graphene nanopores. The simulation details are provided in the revised Fig. 3, Supporting Information Section S2, Fig. S13-S17, Table S1 and S2. We simulated water vapor permeation through 13 graphene nanopores smaller than 10 Å diameter and with carbon, hydrogen, and hydroxyl terminal groups on the pore rim. The relative humidity was set to 55-56% in these simulations, higher than in our experiments, in an effort to promote condensation. Adsorption of water vapor molecules to the graphene was observed and quantified by calculating the pressure drop in the bulk gas due to increased gas molecule density near the surface. However, our simulations show no condensation within the pores or on the graphene. We also calculated the average number of molecules within a cylinder of the pore diameter and

extending 10 Å above and below each pore. The maximum value over all pores was less than 0.03, indicating that there were not prolonged residence times of water molecules near the pore.

We note also that our simplified modeling used the pore size distribution and density measured by STEM with no free parameters used to fit the experimental data. If liquid water formed within the pores, we would expect the transport rates to be significantly altered. The fact that the vapor transport model agrees with the measurements is perhaps evidence that condensation is not occurring in a significant fraction of the pores.

For these reasons, we believe that the water remains in the low density state in the vicinity of the pores.

The Authors confirm the size of the pores in graphene by measuring the transport of KCl ions. This reminded me of simulation results (Konatham et al., Langmuir, many years ago) where ions transport was simulated through graphene pores. Are those results consistent with the interpretation provided here? Related, could free energy barriers such as those extracted from those simulations be useful for informing the engineering models used in the present manuscript?

Konatham et al. [Langmuir 2013, 29, 11884, doi.org/10.1021/la4018695] performed molecular dynamics simulations and reported that pristine (non-functionalized) graphene pores of diameter ~ 7.5 Å can effectively reject NaCl permeation, whereas pristine graphene pores with the diameter ~ 10.5 Å and 14.5 Å easily allow NaCl penetration. This conclusion is consistent with our experimental result in which sub-nanometer graphene pores (~ 2.8 -6.6 Å) effectively reject KCl, NaCl, L-Tr and B12.

Indeed, the energy barrier to crossing graphene nanopores is a critical parameter determining graphene membrane performance [Ref. 38, Nat. Nanotechnol. 2017, 12, 509, doi.org/10.1038/nnano.2017.72]. To achieve high permeance and selectivity, graphene nanopores must have a low energy barrier to permeate molecule crossings and a high energy barrier to the molecules being rejected. High energy barriers occur when the electron clouds of the molecule and carbon atoms on the pore rim overlap appreciably during the crossing. On the other hand, when the pore size is much larger than the size of the crossing molecule, there is essentially no overlap between the electron clouds and the energy barrier drops to zero, independent of pore size beyond a point. The permeation rate through these pores will be much larger than through smaller pores with a high energy barrier and will dominate transport. Density functional theory simulations commonly use energy barrier differences to estimate the selectivity of graphene nanopores to help identify optimal pore geometries for particular separation applications. [Ref. 38, Nat. Nanotechnol. 2017, 12, 509, doi.org/10.1038/nnano.2017.72].

REVIEWERS' COMMENTS

Reviewer #1 (Remarks to the Author):

With the revised manuscript, authors have addressed each of the queries in a satisfactory way by introducing state-of-the-art molecular dynamics simulations in support of their experimental observations. It is quite intriguing to know that the enhancement remain similar (around 3 times) for both the water vapour and liquid water transports by incorporating other influential factors at nanoscale confinements. Authors put all the major queries in a concise way in the revised main draft as well as in supplementary information. Also, authors accepted inherent challenges to address few queries at this moment from experimental point of view.

Now, the revised draft and supplementary file represent solid research work towards authors' findings and claims.

With this, I would recommend this manuscript for publication in Nature Communications journal.

Reviewer #3 (Remarks to the Author):

The comments are sufficiently addressed. I support publication of this work.

Reviewer #4 (Remarks to the Author):

I believe the Authors have considered and adequately addressed all comments from the 4 reviewers. They have conducted additional simulations to confirm the original interpretations, and they put their results in perspective with the existing literature.

In my opinion, the manuscript has been improved significantly, and I recommend publication in its present form.

Reviewer #1 (Remarks to the Author):

With the revised manuscript, authors have addressed each of the queries in a satisfactory way by introducing state-of-the-art molecular dynamics simulations in support of their experimental observations. It is quite intriguing to know that the enhancement remain similar (around 3 times) for both the water vapour and liquid water transports by incorporating other influential factors at nanoscale confinements. Authors put all the major queries in a concise way in the revised main draft as well as in supplementary information. Also, authors accepted inherent challenges to address few queries at this moment from experimental point of view. Now, the revised draft and supplementary file represent solid research work towards authors' findings and claims. With this, I would recommend this manuscript for publication in Nature Communications journal.

We thank the reviewer for recommending publication of our paper.

Reviewer #3 (Remarks to the Author):

The comments are sufficiently addressed. I support publication of this work.

We thank the reviewer for recommending publication of our paper.

Reviewer #4 (Remarks to the Author):

I believe the Authors have considered and adequately addressed all comments from the 4 reviewers. They have conducted additional simulations to confirm the original interpretations, and they put their results in perspective with the existing literature. In my opinion, the manuscript has been improved significantly, and I recommend publication in its present form.

We thank the reviewer for recommending publication of our paper.